# BCQ: BLOCK CLUSTERED QUANTIZATION FOR 4-BIT (W4A4) LLM INFERENCE

## ABSTRACT

Post-training quantization (PTQ) is a promising approach to reducing the storage and computational requirements of large language models (LLMs) without additional training cost. Recent PTQ studies have primarily focused on quantizing only weights to sub-8-bits while maintaining activations at 8-bits or higher. Accurate sub-8-bit quantization for both weights and activations without relying on quantization-aware training remains a significant challenge. We propose a novel quantization method called block clustered quantization (BCQ) wherein each operand tensor is decomposed into blocks (a block is a group of contiguous scalars), blocks are clustered based on their statistics, and a dedicated optimal quantization codebook is designed for each cluster. As a specific embodiment of this approach, we propose a PTQ algorithm called Locally-Optimal BCQ (LO-BCQ) that iterates between the steps of block clustering and codebook design to greedily minimize the quantization mean squared error. When weight and activation scalars are encoded to W4A4 format (with $0.5$-bits of overhead for storing scaling factors and codebook selectors), we advance the current state-of-the-art by demonstrating $< 1\%$ loss in inference accuracy across several LLMs and downstream tasks.

## 1 INTRODUCTION

Quantization is a highly effective and widely adopted technique for reducing the computational and storage demands of Large Language Model (LLM) inference. While recent efforts (Wang et al., 2023; Tseng et al., 2024; Egiazarian et al., 2024; Frantar et al., 2023; Lin et al., 2023) have largely focused on weight-only quantization targeting single-batch inference, activation quantization becomes critical for improving throughput during multi-batch inference scenarios such as cloud-scale deployments serving multiple users. Previous works (Yao et al., 2023; Dai et al., 2021) on sub-8-bit quantization of both weights and activations have relied on quantization-aware training (QAT) to recover accuracy loss during inference. However, the prohibitive cost of training and unavailability of training data in recent LLMs has made QAT increasingly difficult and motivated recent post-training quantization (PTQ) efforts (Xiao et al., 2023; Rouhani et al., 2023a; Wu et al., 2023).

Block quantization techniques where each block, typically consisting of 16-to-32-scalar elements, has its own scaling factor (Rouhani et al., 2023a; Dai et al., 2021) achieve the current state-of-the-art accuracy for sub-8-bit quantization of both weights and activations. While these works deploy the same quantizer (number-format) across blocks, we hypothesize that one way to achieve lower quantization mean squared error (MSE) would be to design a dedicated codebook for each block through an MSE-optimal algorithm such as Lloyd (1982). However, such a design would be expensive in terms of computational effort and memory footprint. Therefore, we propose to amortize this cost via codebook sharing among clusters of blocks. Our method is called block clustered quantization (BCQ) and is comprised of two steps: (1) a clustering step applied to operand blocks, and (2) a quantization step individually applied to operand scalars based on their cluster membership.

We propose an iterative PTQ algorithm called LO-BCQ (locally optimal block clustered quantization) that jointly optimizes the block clustering and the per-cluster codebook. We prove that LO-BCQ greedily minimizes quantization MSE across iterations by performing locally optimal steps at each iteration. With the optimal codebooks derived through LO-BCQ, we demonstrate state-of-the-art bitwidth-vs-accuracy across a suite of GPT3, Llama2 and Nemotron-4 models on a wide range of downstream tasks. For all our results, we employ PTQ on frozen model parameters.

## 1.1 RELATED WORK

Recent sub-4-bit quantization proposals such as (Wang et al., 2023; Tseng et al., 2024; Egiazarian et al., 2024) explore extreme weight quantization while maintaining activations at 8-bit or higher precision. In particular, BitNet (Wang et al., 2023) proposed W1A8 quantization resulting in an aggregate (weights + activations) bitwidth comparable to LO-BCQ. However, BitNet demands training from scratch and despite this large training cost suffers significant loss in accuracy in downstream tasks. QuiP# (Tseng et al., 2024) and AQLM (Egiazarian et al., 2024) propose W2A8 quantization through codebooks. These methods explore vector and additive codebook quantization, respectively, and rely on a significantly large number of codebooks (of the order $2^{16}$) for quantization, suffering large decoding costs. In contrast, LO-BCQ explores scalar quantization methods for W4A4 quantization and achieves $< 1\%$ accuracy loss in downstream tasks with no more than 16 codebooks with 16 entries each. W4A8 quantization has been proposed in (Frantar et al., 2023; Bai et al., 2021; Yao et al., 2022) involving weight updates to preserve accuracy and in (Lin et al., 2023; van Baalen et al., 2024) without any weight updates (PTQ). Further, (Guo et al., 2023; Wei et al., 2023; Kim et al., 2023a) perform sub-8-bit weight quantization by suppressing outliers. In contrast, LO-BCQ explores sub-8-bit activation quantization alongside weight quantization.

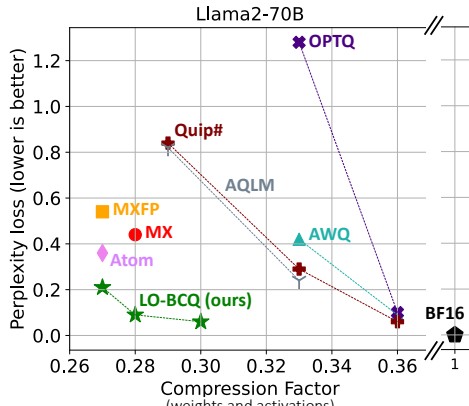

Figure 1: Wikitext perplexity loss relative to unquantized baseline vs compression factor of LO-BCQ compared to previous LLM quantization proposals. Here, compression factor is the cumulative number of bits in the weight and activation tensors[1] that need to be processed in each layer relative to an unquantized BF16 baseline.

Block quantization has emerged as an effective technique for quantizing both weights and activations, as demonstrated in VSQ (Dai et al., 2021), FineQuant (Kim et al., 2023b), ZeroQuant-V2 (Yao et al., 2023), Atom (Zhao et al., 2024) through integer number formats, and in (Zhang et al., 2023), ZeroQuant-FP (Wu et al., 2023), MX (Rouhani et al., 2023a) and MXFP (Rouhani et al., 2023b) through floating-point formats. Moreover, sub-block scaling techniques explored in MXFP and BSFP (Lo et al., 2023) demonstrate improvements over standard block quantization. In this work, we perform clustering of operand blocks and share MSE-optimal codebook quantizers among the scalars of each cluster. Minimizing quantization MSE using the 1D (Lloyd-Max) and 2D Kmeans clustering has been explored in (Han et al., 2016; Cho et al., 2021; 2023) and (van Baalen et al., 2024), respectively. In contrast, LO-BCQ iteratively optimizes block clustering alongside Lloyd-Max based optimal scalar quantization of block clusters.

Figure 1 compares the perplexity loss vs compression factor of LO-BCQ to other quantization proposals. Here, the perplexity loss is relative to an unquantized baseline on the Wikitext-103 dataset for LO-BCQ, MX and MXFP4, and on the Wikitext2 for others. Further, the compression factor refers to the total number of bits in the weight and activation[1] tensors (computed as $|A|B_A + |W|B_W$ following Sakr et al. (2017))[2] that need to be processed in each layer relative to an unquantized BF16 baseline. Depending on the target application, weight or activation quantization may be more important. For the sake of generality, we consider them to be equally important in our metric. As shown in Figure 1, LO-BCQ advances the current state-of-the-art by achieving the best trade-off between perplexity and compression.

## 1.2 CONTRIBUTIONS

The main contributions of this work are as follows:

---

[1]The size of activations is measured for the prefill phase with a context length of 4096 and batch size of 1.

[2]the notation $|X|$ refers to the total number of scalars in tensor $X$, and $B_X$ is the bitwidth of $X$.

- We propose BCQ, a block clustered quantization framework that performs per-block quantization by first clustering operand blocks and then quantizing each block cluster using a dedicated codebook.
- We derive a locally optimal version of BCQ called LO-BCQ that iteratively optimizes block clustering and per-cluster quantization to provably minimize quantization MSE for any value distribution. We demonstrate that LO-BCQ is applicable to quantization of both weights and activations of LLMs.
- We propose block formats for LO-BCQ where each operand block is associated with an index that maps it to one of a set of codebooks, and a group of blocks (called a block array) share a quantization scale-factor. We vary the length of blocks, block arrays and the number of codebooks to study different configurations of LO-BCQ.
- When each of the weight and activation scalars are quantized to 4-bits (effective bitwidth including per-block scale-factors etc. is $4.5$ to $4.625$ bits), we achieve $< 0.1$ loss in perplexity across GPT3 (1.3B, 8B and 22B) and Llama2 (7B and 70B) models and $< 0.2$ loss in the Nemotron4-15B model, respectively, on the Wikitext-103 dataset. Further, we achieve $< 1\%$ loss in average accuracy across downstream tasks such as MMLU and LM evaluation harness.

To the best of our knowledge, we are the first to achieve $< 1\%$ loss in downstream task accuracy when both LLM activations and weights are quantized to 4-bits during PTQ ( no finetuning).

## 2 BLOCK CLUSTERED QUANTIAZTION (BCQ)

In this section, we introduce the concept of block clustered quantization (BCQ) and present the locally optimal block clustered quantization (LO-BCQ) algorithm that minimizes quantization MSE for any operand. We also introduce block formats to support various LO-BCQ configurations.

### 2.1 MATHEMATICAL DEFINITION

Given a tensor $X$ composed of $L_X$ scalar elements, we denote its blockwise decomposition as $\{b_i\}_{i=1}^{N_b}$, where $b_i$'s are blocks of $L_b$ consecutive elements in $X$, and the number of blocks is given by $N_b = L_X/L_b$. Block clustered quantization (see Figure 2) uses a family of $N_c$ codebooks $\mathcal{C} = \{C_i\}_{i=1}^{N_c}$, where $N_c << N_b$, and clusters the blocks into $N_c$ clusters such that each is associated with one of the $N_c$ codebooks. This procedure is equivalent to creating a mapping function $f$ from a block $b$ to a cluster index in $\{1, \ldots, N_c\}$. Quantization (or encoding) proceeds in a two-step process: (i) *mapping* to assign a cluster index to a given block, and (ii) *quantization* of its scalars using the codebook corresponding to that index. Formally, denoting $\hat{b}$ as the result of block clustered quantization of a given block $b$ in $X$, this procedure is described as:

$$\hat{b} = C_{f(b)}(b) \tag{1}$$

where $C$ is a $2^B$-entry codebook that maps each scalar in $b$ to a $B$-bit index to the closest representation. Each quantized scalar of block $b$ is obtained as:

$$\hat{b}[l] = \arg \min_{k=1\ldots 2^B} |b[l] - C_{f(b)}[k]|^2 \tag{2}$$

where the notation $x[y]$ is used to describe the $y^{\text{th}}$ element in an arbitrary block $x$. That is, each scalar in $\hat{b}$ is an index to the closest entry by value in $C_{f(b)}$.

Once mapped by invoking $f$, we store the $log2(N_c)$-bit codebook selector for each block. Therefore, the effective bit-width of each quantized scalar is given by:

$$\text{Bitwidth}_{\text{BCQ}} = B + log2(N_c)/L_b \tag{3}$$

### 2.2 LOCALLY OPTIMAL BLOCK CLUSTERED QUANTIZATION (LO-BCQ)

Our goal is to construct a family of codebooks $\mathcal{C}$ resulting in minimal quantization MSE during block clustered quantization. Figure 3 presents an algorithm called Locally Optimal BCQ (LO-BCQ) to achieve this goal. LO-BCQ consists of two main steps: (i) updating block clusters with fixed per-cluster codebooks, and (ii) updating per-cluster codebooks with fixed block clusters. This algorithm begins at iteration 0 (initial condition) with a set of $N_c$ initial codebooks $\{C_1^{(0)}, \ldots, C_{N_c}^{(0)}\}$ and unquantized operand blocks as inputs. During step 1 of iteration $n$, with the per-cluster codebooks

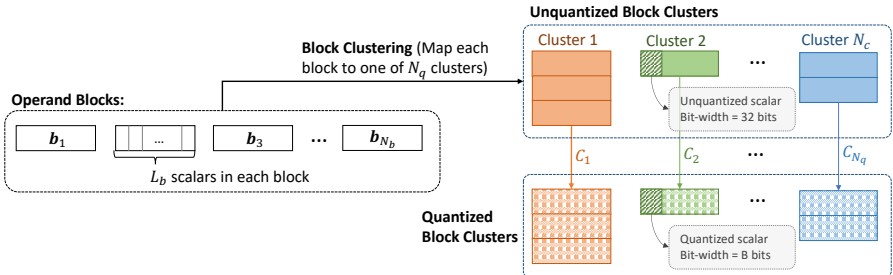

Figure 2: Block clustered quantization: Each operand block is first mapped to a cluster based on a mapping function and then each scalar of that block is encoded as a $B$-bit index to the closest entry in the $2^B$-entry codebook associated with that cluster.

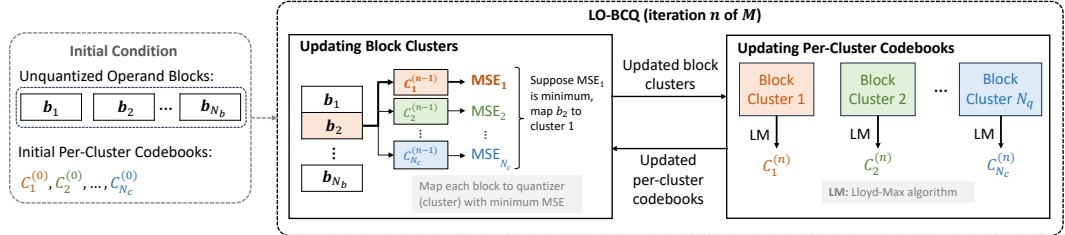

Figure 3: Overview of LO-BCQ algorithm: The algorithm starts with a set of initial per-cluster codebooks, and then iteratively performs two steps (i) fix per-cluster codebooks and update block clusters and (ii) fix block clusters and update per-cluster codebooks.

from the previous iteration $\{C_1^{(n-1)}, \dots C_{N_c}^{(n-1)}\}$, we perform block clustering by mapping each block to the codebook that achieves minimum quantization MSE. That is, we use the following mapping function:

$$f^{(n)}(\boldsymbol{b}) = \arg\min_{i=1\dots N_c} \|\boldsymbol{b} - C_i^{(n-1)}(\boldsymbol{b})\|_2^2 \tag{4}$$

Since each codebook $C_i$ is associated with a cluster $i$, mapping to $C_i$ is equivalent to mapping to cluster $i$. Specifically, at iteration $n$, we construct $N_c$ block clusters $\boldsymbol{\mathcal{B}}^{(n)} = \{\mathcal{B}_i^{(n)}\}_{i=1}^{N_c}$, where each cluster is defined as:

$$\mathcal{B}_i^{(n)} = \left\{\boldsymbol{b}_j \big| f^{(n)}(\boldsymbol{b}_j) = i \text{ for } j \in \{1\dots N_b\}\right\} \tag{5}$$

In step 2, given the updated block clusters from step 1 and quantization bitwidth $B$, we apply the Lloyd-Max algorithm on each block cluster to derive optimal $2^B$-entry per-cluster codebooks $\{C_1^{(n)}, \dots C_{N_c}^{(n)}\}$:

$$C_i^{(n)} \leftarrow \text{LloydMax}(\mathcal{B}_i^{(n)}, B) \tag{6}$$

where the Lloyd-Max algorithm (see A.1, Lloyd-Max is equivalent to K-means clustering on 1-dimensional data) is invoked on the data of the corresponding cluster $\mathcal{B}_i^{(n)}$.

We iterate steps 1 and 2 until convergence or a pre-determined number of iterations $M$. Empirically, we find that LO-BCQ converges at $M <= 100$. Since each of these steps are locally optimal, we find that the quantization MSE is non-increasing for each iteration. As a result, for any given value distribution, our LO-BCQ algorithm greedily minimizes quantization MSE. A theoretical proof of this claim is provided in section A.2.

## 2.3 Convergence and Initialization

Prior to clustering, we find that normalizing the operand blocks improves convergence. However, a block-wise normalization factor (or scaling factor) induces computational and memory overheads. Therefore, we perform a second-level quantization of this scaling factor to $B_s$-bits and share it across

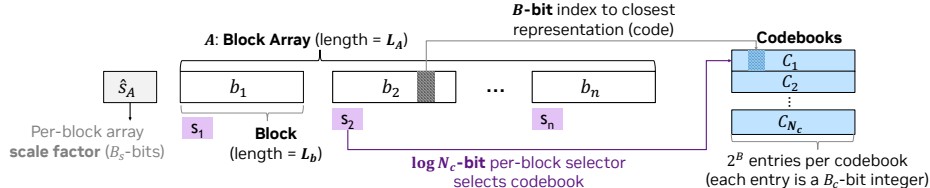

Figure 4: Block format for LO-BCQ. Each operand block is associated with a $log2(N_c)$-bit selector that selects the best codebook and each scalar is a $B$-bit index that represents the closest value in the selected codebook. Each block array $\boldsymbol{A}$ is associated with a $B_s$-bit scale factor.

an array of blocks of length $L_A$. Furthermore, better convergence is observed for larger number of codebooks ($N_c$) and for a smaller block length ($L_b$). Such trends increase the bitwidth of BCQ in equation 3, meaning that LO-BCQ has an inherent trade-off between accuracy and complexity.

We initialize the per-cluster codebooks $\{C_1^{(0)}, \ldots, C_{N_c}^{(0)}\}$ based on K-means++ initialization algorithm which maximizes pairwise euclidean distances. In our experiments, we found such initialization to lead to significantly better convergence than a random one. Further, in step 2 of each iteration, when Lloyd-Max algorithm is invoked in equation 6, we set the initial centroids corresponding to the codebooks identified in the previous iteration.

### 2.4 BLOCK FORMATS FOR LO-BCQ

Figure 4 illustrates the LO-BCQ block format where each operand block of length $L_b$ is associated with a $log2(N_c)$-bit index (result of the mapping function $f$ in 4) that selects the best codebook for that block. Each codebook is composed of $2^B$ entries and each scalar in the operand block is a $B$-bit index that represents the closest value in the selected codebook. Each entry in the codebook is a $B_c$-bit integer ($B_c > B$). Finally, each block array $\boldsymbol{A}$ is associated with a scale-factor $s_A$. This scale-factor and its quantization $\hat{s}_A$ to $B_s$-bits are computed as:

$$s_A = \left(2^{B_c-1} - 1\right)/\max(\mathrm{abs}(\boldsymbol{A})); \quad \hat{s}_A = Q_F\{s_A/s_X, B_s\} \tag{7}$$

where $s_X$ is a per-tensor scale-factor that is shared by the entire operand tensor $\boldsymbol{X}$ and $Q_F$ denotes a quantizer that quantizes a given operand to format $F$ (see section A.4 for more details on number formats and quantization method).

The bitwidth of LO-BCQ is computed as:

$$\mathrm{Bitwidth}_{\mathrm{LO-BCQ}} = B + log2(N_c)/L_b + B_s/L_A + N_c * 2^B * B_c/L_X \tag{8}$$

where the term $N_c * 2^B * B_c/L_X$ is usually negligible since the memory footprint of codebooks (numerator) is negligible compared to the size of the operand tensor (denominator). Indeed, we emphasize that LO-BCQ shares a set of $N_c <= 16$ codebooks among the scalars of the entire tensor, resulting in negligible memory overhead for storing the codebooks.

Table 1: Various LO-BCQ configurations and their bitwidths.

| $N_c$ \ $L_A$ | $L_b = 8$ | | | | $L_b = 8$ | | $L_b = 2$ |
|---|---|---|---|---|---|---|---|
| | 2 | 4 | 8 | 16 | 2 | 4 | 2 |
| 64 | 4.25 | 4.375 | 4.5 | 4.625 | 4.375 | 4.625 | 4.625 |
| 32 | 4.375 | 4.5 | 4.625 | 4.75 | 4.5 | 4.75 | 4.75 |
| 16 | 4.625 | 4.75 | 4.875 | 5 | 4.75 | 5 | 5 |

In this paper, we assume $B_s = 8$ and the data format $F$ is floating point E4M3. Further, each codebook entry is a 6-bit integer (i.e, $B_c = 6$) and we vary $N_c$ between 2 and 16, $L_b$ between 2 and 8, and $L_A$ between 16 and 64 to obtain various LO-BCQ configurations. We list the configurations and their corresponding bitwidths in Table 1.

Figure 5 compares our 4-bit LO-BCQ block format to MX (Rouhani et al., 2023a). As shown, both LO-BCQ and MX decompose a given operand tensor into block arrays and each block array into blocks. Similar to MX, we find that per-block quantization ($L_b < L_A$) leads to better accuracy due to increased flexibility. While MX achieves this through per-block 1-bit micro-scales, we associate a dedicated codebook to each block through a per-block codebook selector. Further, MX quantizes the per-block array scale-factor to E8M0 format without per-tensor scaling. In contrast during LO-BCQ, we find that per-tensor scaling combined with quantization of per-block array scale-factor to E4M3 format results in superior inference accuracy across models.

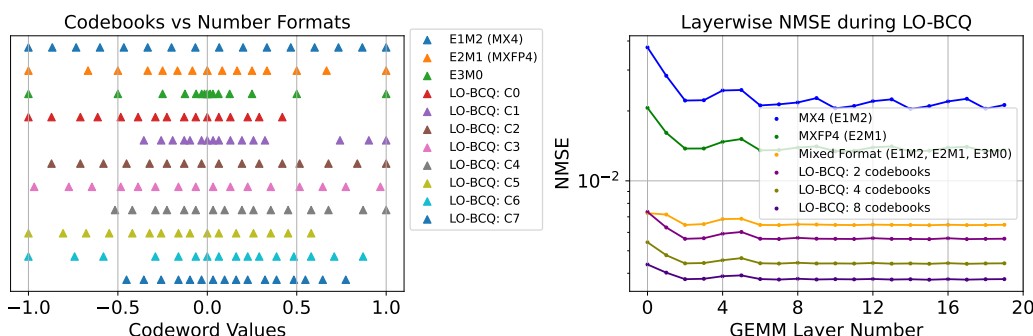

Figure 5: Comparing LO-BCQ to MX format.

Figure 6: LO-VCQ codebooks compared to 4-bit floating point formats and layerwise normalized MSE (NMSE). We compute NMSE for the weights of first 20 GEMM layers (QKV, projection and fully-connected) of Llama2-7B model. Note that we use the NMSE for better visualization across varying layer data.

## 3 PRACTICAL IMPLEMENTATION OF LO-BCQ FOR LLM INFERENCE

In this section, we discuss specifics of a practical implementation of LO-BCQ for inference. Specifically, we first describe the codebook design process, followed by the practical mechanism for activation quantization on-the-fly.

We pre-calibrate the LO-BCQ codebooks for both weights and activations offline (prior to inference). Since weights are known, their own data can be used as calibration set. On the other hand, activations are dynamic and vary for every input; thus, as per common quantization strategies (Wu et al., 2020a; Sakr et al., 2022), we employ a randomly sampled calibration set from training data in order to build activation codebooks. Once codebooks are calibrated, we also quantize the codewords to 6-bit integers to further improve the energy efficiency of GEMM hardware. The choice of 6-bit was based on empirical observations of accuracy being maintained with $L_A <= 64$.

Figure 6 compares the codebooks identified by the LO-BCQ algorithm in a GEMM layer of a GPT3-126M model to 4-bit floating point formats such as E1M2,

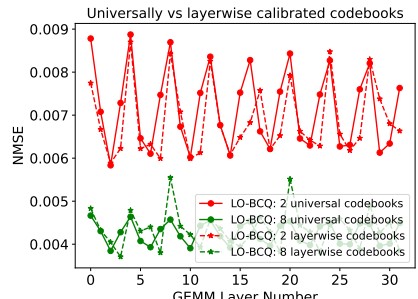

Figure 7: Quantization NMSE acheived by universally calibrated codebooks compared to that calibrated layerwise in Llama2-7B inputs of first 30 GEMM (QKV, projection and fully-connected) layers.

E2M1 and E3M0. As shown, the LO-BCQ codebooks outperform other block formats by capturing the arbitrary and non-uniform patterns in the value distributions of LLM operands and allowing each block to map to the codebook that best represents it. The mapping of operand blocks to the best of available codebooks can be conceptually compared to prior works that have explored mixed-format quantization such as (Tambe et al., 2020; Zadeh et al., 2022).

LO-BCQ provides the quantization operation the flexibility to assign data to any of the sign posts (codewords) in Figure 6. The union of these sign posts covers the real line with a resolution that is clearly superior to that of a 4-bit quantizer. Therefore, we hypothesized that these codebooks need not be calibrated on a per-tensor (layerwise) basis, but rather, it is likely that they would be universally appropriate to quantize *any tensor, at any layer, for any model*. To verify this hypothesis, we calibrated a set of codebooks on data sampled from GPT3 models on Wikitext-103 dataset and froze it. We find that these codebooks achieve comparable quantization MSE compared to those calibrated individually on each operand as shown in Figure 7 which verifies our hypothesis. In our subsequent results, we always employ universally calibrated codebooks.

Finally, we note that in a real implementation, activations can be efficiently quantized on the fly. Indeed, LO-BCQ involves computing the following values – per-block array scale-factor $s_A$, per-block codebook selector $s_b$ which is the result of the mapping function $f$ (Eq. 4), and the index to closest representation $\hat{b}$ in the selected codebook (Eq. 2). Note that the computation of $s_A$ simply corresponds to a max-reduction (followed by quantization) over the block array, whose size is small ($<= 64$). Importantly, with LO-BCQ, the size of codebooks ($<= 0.19KB$) is small enough such that it easily fits within the shared memory of modern GPUs. This is an important distinction with other works on codebook quantization (Tseng et al., 2024; Egiazarian et al., 2024). As such, $s_b$ and $\hat{b}$ can be concurrently computed in a thread-local sub-routine within a custom CUDA kernel. The locality of computation circumvents the need for any synchronization of streaming multiprocessors.

## 4 EXPERIMENTAL EVALUATION OF LO-BCQ

In this section, we present our accuracy studies on downstream tasks comparing LO-BCQ to various other block quantization proposals. Next, we present ablation studies on varying LO-BCQ configurations and our calibration methodology, namely universal vs local.

### 4.1 EXPERIMENTAL SETUP

We perform accuracy studies on GPT3 (Shoeybi et al., 2020) (1.3B, 8B and 22B), Llama2 (Touvron et al., 2023) (7B and 70B) and Nemotron4-15B (Parmar et al., 2024) models. We evaluate PTQ inference accuracy on several downstream tasks including Wikitext-103 (Merity et al., 2016), MMLU (Hendrycks et al., 2021) and Eleuther AI's LM evaluation harness (Gao et al., 2024). In LM evaluation harness, we infer on Race (RA), Boolq (BQ), Hellaswag (HS), Piqa (PQ) and Winograde (WG) tasks and in the MMLU dataset we evaluate all tasks. In all these models, we quantize GEMM layers including Query, Key and Value computations, Projection layer after self attention and the fully-connected layers.

We apply the LO-BCQ algorithm to the operands before inference and pre-calibrate the optimal codebooks. In our experiments, we perform this calibration on one batch of activations from the training data of the GPT3-126M model and the Wikitext-103 dataset. We freeze these optimal codebooks across operands and models during all of our accuracy evaluations. Further, we represent each entry of the codebooks as a 6-bit integer. That is, once decoded, the inner product computations with a block array during inference can be performed at 6-bit precision[3]. Furthermore, we perform ablation studies on the LO-BCQ configurations listed in Table1 with quantization bitwidth ranging from 4.25-bits to 5-bits. We denote the LO-BCQ configurations by the tuple $\{L_A, L_b, N_c\}$.

We compare LO-BCQ against previous block quantization works that have explored PTQ of both weights and activations such as VSQ (Dai et al., 2021), MX (Rouhani et al., 2023a) and MXFP (Rouhani et al., 2023b). VSQ and MX perform per-block quantization of 16-element blocks with an 8-bit scale-factor per-block resulting in an effective bit-width of 4.5 bits. VSQ quantizes each scalar to INT4 format and per-block scale-factor to INT8 format. MX performs micro-scaling at per-block level with a 1-bit exponent shared by 2-element blocks. Each scalar is quantized to INT3. In this paper, we overestimate accuracy of MX by allowing each scalar to have its own exponent, resulting in INT4 precision. The per-block array scale factors of MX are quantized to E8M0 format. Therefore, our evaluation results in a bitwidth of 4.5 bits. Further, MXFP explores 32-element blocks with 8-bit scale-factor per block resulting in an effective bitwidth of 4.25 bits. The number format of scalars and per-block scale factors are E2M1 and E8M0, respectively. The quantization methodology with these block formats is detailed in A.4.4.

---

[3]In our experiments in this paper, we emulate ("fake") quantization by representing the quantized values in BF16 format. Therefore, the computations are performed in BF16 precision.

Additionally, we compare weight-only (W4A8) LO-BCQ to other weight-only quantization proposals of equivalent bitwidth such as GPTQ (Frantar et al., 2023), AWQ (Lin et al., 2023), OmniQ (Shao et al., 2024) and QuiP# (Tseng et al., 2024). For this comparison, we choose a block-array length of $128$ for LO-BCQ, matching the group-size of other works.

## 4.2 Accuracy studies on downstream tasks

Table 2 presents our comprehensive accuracy evaluations across the Llama2 and GPT3 models, on the Wikitext-103, LM evaluation harness and MMLU datasets. For convenience, we present select LO-BCQ configurations in this table. See A for accuracy studies on other configurations.

### 4.2.1 Perplexity on Wikitext-103

Across large models such as Llama2-70B and GPT3-22B, $4.5$-bit LO-BCQ achieves $< 0.1$ loss in perplexity compared to the unquantized baseline on the Wikitext-103 dataset. Further, LO-BCQ achieves significant benefits compared to the baselines of equivalent bit-widths. When the quantization bitwidth is $4.5$-bits, LO-BCQ achieves an average improvement of $0.9$ and $0.76$ in perplexity compared to VSQ and MX, respectively, and $1.19$ average improvement with $4.25$-bits compared to MXFP across models. We achieve these improvements during PTQ, i.e., without any additional training or finetuning.

MX, MXFP and VSQ perform per-block quantization by associating a scale-factor to each block (or a block array) and with a single number format (quantizer) across blocks. On the other hand, in addition to per-block array scaling, LO-BCQ allows a block to flexibly map to a codebook that best represents it from a set of codebooks. This flexibility allows LO-BCQ to achieve better perplexity. Furthermore, we find that with a larger quantization bitwidth, LO-BCQ achieves better perplexity across models as expected.

Further, the number format of per-block (or block array) scale-factor has a significant impact on accuracy. VSQ is unable to sufficiently capture the range of activations with its INT8 scale-factors as observed in Llama2-7B, while it outperforms the E8M0 scale-factors of MX in GPT3-22B due to better resolution when representing large values. Across various models, we find that the E4M3 format of LO-BCQ provides sufficient range and resolution to represent the scale-factors.

### 4.2.2 Accuracy on LM evaluation harness tasks

Across $0$-shot LM evaluation harness tasks LO-BCQ shows significant improvement in average accuracy compared to MX, MXFP and VSQ at equivalent bitwidth. Further, across models during $4.5$-bit quantization, LO-BCQ achieves $< 1\%$ loss in average accuracy compared to the respective unquantized baselines. When the bitwidth of LO-BCQ is increased by varying its configuration, we find that the average accuracy generally increases albeit with a few exceptions. Although these variations are small ($< 0.5\%$), we believe that they arise due to the universal calibration of codebooks. Our codebooks are calibrated on a batch of training data from the Wikitext-103 dataset and the GPT3-126M model and remain frozen across all datasets and models.

### 4.2.3 Accuracy on MMLU tasks

Similarly, in 5-shot MMLU tasks LO-BCQ achieves $< 1\%$ loss in average accuracy with $4.5$-bits per scalar compared to respective unquantized baselines across GPT3-22B and Llama2-70B models. Further, LO-BCQ achieves a significantly better accuracy compared to all of our block quantization baselines such as VSQ, MX and MXFP4 at equivalent bitwidth. Across Llama2 models, LO-BCQ with a smaller bitwidth ($4.25$-bits) outperforms VSQ and MX4 with a comparatively larger bitwidth ($4.5$-bits). While the $0.5$-bit overhead in VSQ and MX4 are used on per-block array scale-factors, the $0.25$-bit overhead of LO-BCQ is shared between scale-factors and codebook selectors. Therefore, the superior accuracy of LO-BCQ can be attributed to the better representation by selecting the best codebook for each block.

### 4.2.4 Accuracy studies on Nemotron4-15B

Table 3a lists the perplexity achieved by the Nemotron4-15B model quantized by LO-BCQ on Wikitext-103 dataset and compares it to various baselines. When both weights and activations are quantized to $4.75$-bits, LO-BCQ achieves $0.16$ loss in perplexity compared to unquantized baseline.

Table 2: PTQ Perplexity (lower is better) on Wikitext-103 dataset and downstream task accuracy (higher is better) with Llama2 and GPT3 models. We denote the LO-BCQ configurations by the tuple $\{L_A, L_b, N_c\}$ = {Length of block array, Length of block, Number of codebooks}.

| Method | Bitwidth | Wiki3 | LM evaluation Harness (Accuracy %, 0-shot) | | | | | | MMLU (5-shot) |
|---|---|---|---|---|---|---|---|---|---|
| | | PPL ($\Delta$) | RA | BQ | WG | PQ | HS | Avg ($\Delta$ %) | Avg ($\Delta$ %) |
| Llama2-7B | | | | | | | | | |
| FP32 | 32 | 5.06 | 44.4 | 79.29 | 69.38 | 78.07 | 57.10 | 65.65 | 45.8 |
| MX4 | 4.5 | 5.73 (0.67) | 41.43 | 73.98 | 66.22 | 77.04 | 55.19 | 62.77 (2.88) | 41.38 (4.42) |
| VSQ | 4.5 | 835 (829) | 31.39 | 65.75 | 55.49 | 67.30 | 43.51 | 52.69 (12.96) | 26.48 (19.3) |
| MXFP4 | 4.25 | 5.76 (0.70) | 41.34 | 74.00 | 67.48 | **77.53** | 54.22 | 62.91 (2.74) | 37.64 (8.16) |
| **LO-BCQ** $\{64, 8, 2\}$ | 4.25 | 5.31 (0.25) | 42.49 | 77.58 | **68.90** | 77.09 | 55.93 | 64.40 (1.25) | 43.90 (1.90) |
| **LO-BCQ** $\{64, 8, 8\}$ | 4.5 | 5.19 (0.13) | 42.58 | 77.43 | **69.77** | 77.09 | 56.51 | 64.68 (0.97) | 43.90 (1.90) |
| **LO-BCQ** $\{32, 8, 16\}$ | 4.75 | **5.15 (0.09)** | **43.73** | 77.86 | **68.90** | 77.86 | 56.52 | **64.97 (0.68)** | 44.50 (1.30) |
| Llama2-70B | | | | | | | | | |
| FP32 | 32 | 3.14 | 48.8 | 85.23 | 79.95 | 81.56 | 65.27 | 72.16 | 69.12 |
| MX4 | 4.5 | 3.58 (0.44) | **48.04** | 82.41 | 76.40 | **80.58** | 63.24 | 70.13 (2.03) | 65.73 (3.39) |
| VSQ | 4.5 | 4.96 (1.82) | 47.85 | 82.29 | 77.27 | 79.82 | 61.40 | 69.73 (2.43) | 62.46 (6.66) |
| MXFP4 | 4.25 | 3.69 (0.55) | 47.75 | 83.06 | 76.32 | **80.58** | 63.24 | 70.19 (1.97) | 66.16 (2.96) |
| **LO-BCQ** $\{64, 8, 2\}$ | 4.25 | 3.35 (0.21) | **49.0** | 82.82 | 78.77 | 81.45 | 64.21 | 71.25 (0.91) | 68.07 (1.05) |
| **LO-BCQ** $\{64, 8, 8\}$ | 4.5 | **3.23 (0.09)** | 49.28 | 84.03 | 78.37 | 81.45 | 64.76 | 71.58 (0.58) | **68.17 (0.95)** |
| **LO-BCQ** $\{32, 8, 16\}$ | 4.75 | **3.20 (0.06)** | 49.28 | 84.93 | 80.66 | 81.34 | 65.18 | **72.28 (+0.12)** | 68.27 (0.85) |
| GPT3-1.3B | | | | | | | | | |
| FP32 | 32 | 9.98 | 37.51 | 64.62 | 58.01 | 74.21 | 43.51 | 55.57 | 24.20 |
| MX4 | 4.5 | 11.33 (1.35) | 35.22 | 54.31 | **57.38** | 70.78 | 40.58 | 51.65 (3.92) | 24.04 |
| VSQ | 4.5 | 10.83 (0.85) | 35.98 | 62.60 | **59.59** | 71.27 | 39.98 | 53.88 (1.69) | 25.89 |
| MXFP4 | 4.25 | 11.04 (1.06) | **36.56** | 61.68 | 56.75 | 71.65 | 40.66 | 53.46 (2.11) | 24.87 |
| **LO-BCQ** $\{64, 8, 2\}$ | 4.25 | 10.40 (0.42) | **36.94** | 63.73 | **58.17** | 73.01 | 42.10 | 54.79 (0.78) | 24.80 |
| **LO-BCQ** $\{64, 8, 8\}$ | 4.5 | 10.17 (0.19) | 36.27 | 63.49 | **57.85** | 73.07 | **42.73** | 54.68 (0.89) | 24.50 |
| **LO-BCQ** $\{32, 8, 16\}$ | 4.75 | 10.12 (0.14) | **37.03** | 63.33 | **58.56** | 73.94 | 43.20 | 55.07 (0.50) | 24.80 |
| GPT3-8B | | | | | | | | | |
| FP32 | 32 | 7.38 | 41.34 | 68.32 | 67.88 | 78.78 | 54.16 | 62.10 | 25.50 |
| MX4 | 4.5 | 8.15 (0.77) | 38.28 | 66.27 | 65.11 | 75.63 | 50.77 | 59.21 (2.89) | 24.51 |
| VSQ | 4.5 | 8.17 (0.79) | **40.86** | 63.91 | 66.93 | 76.28 | 51.38 | 59.87 (2.23) | 27.57 |
| MXFP4 | 4.25 | 9.12 (1.74) | 39.71 | 65.35 | 67.01 | 76.12 | 50.22 | 59.68 (2.42) | 24.93 |
| **LO-BCQ** $\{64, 8, 2\}$ | 4.25 | 7.61 (0.23) | **40.48** | 69.20 | 66.85 | 77.31 | 53.06 | 61.38 (0.72) | 24.53 |
| **LO-BCQ** $\{64, 8, 8\}$ | 4.5 | **7.48 (0.1)** | 39.43 | 69.45 | 67.72 | 77.75 | 53.71 | 61.61 (0.49) | 26.04 |
| **LO-BCQ** $\{32, 8, 16\}$ | 4.75 | **7.45 (0.07)** | 39.62 | 69.30 | 67.00 | 77.37 | 53.51 | 61.36 (0.74) | 25.32 |
| GPT3-22B | | | | | | | | | |
| FP32 | 32 | 6.54 | 40.67 | 76.54 | 70.64 | 79.16 | 57.11 | 64.82 | 38.75 |
| MX4 | 4.5 | 7.69 (1.15) | 39.04 | 72.26 | 67.96 | 77.86 | 54.77 | 62.38 (2.44) | 37.07 (1.68) |
| VSQ | 4.5 | 7.12 (0.58) | **40.57** | 65.81 | **69.61** | 77.20 | 54.82 | 61.60 (3.22) | **37.79 (0.96)** |
| MXFP4 | 4.25 | 10.18 (3.64) | 39.14 | 69.61 | 64.17 | 75.68 | 47.60 | 59.24 (5.58) | 32.26 (6.49) |
| **LO-BCQ** $\{64, 8, 2\}$ | 4.25 | 6.74 (0.20) | **40.48** | 75.41 | 69.14 | **78.24** | 56.06 | **63.87 (0.95)** | 36.71 (2.04) |
| **LO-BCQ** $\{64, 8, 8\}$ | 4.5 | **6.62 (0.08)** | 39.43 | 77.09 | 70.17 | 78.62 | 56.60 | 64.38 (0.44) | **38.13 (0.62)** |
| **LO-BCQ** $\{32, 8, 16\}$ | 4.75 | **6.59 (0.05)** | 39.62 | 75.35 | 69.30 | 78.89 | 56.64 | 63.96 (0.86) | **38.34 (0.41)** |

Compared to GPT3 and Llama2 models, LO-BCQ suffers a larger perplexity degradation in this model. A similar trend is observed for our block quantization baselines VSQ, MX and MXFP. At equivalent bitwidth LO-BCQ achieves 1.45, 2.75 and 1.94 improvement in perplexity over VSQ, MX and MXFP, respectively.

Across MMLU tasks, LO-BCQ achieves $< 1\%$ loss in average accuracy compared to unquantized baseline with $>= 4.5$-bits per scalar. Further, we achieve 5.57%, 6.34% and 4.89% improvement over MX4, VSQ and MXFP4, respectively, at equivalent bitwidth.

Table 3b compares weight-only (W4A8) LO-BCQ with a block array size of 128 to other weight-only quantization proposals of comparable block array size and effective bit-width. As shown, LO-BCQ with 2, 4, 8 and 16 codebooks with effective bitwidth of 4.19, 4.31, 4.44 and 4.56, respectively, achieves significantly lower perplexity loss. It is worth noting that we evaluate this loss on Wikitext-103 dataset, which is a much larger dataset compared to Wikitext2 used by other works.

## 4.3 ABLATION STUDIES

Table 4a shows the perplexity of LO-BCQ on Wikitext-103 dataset and across Llama2-70B and GPT3-22B models when its configuration is varied. For a given $L_b$ (block length), larger number of codebooks results in better perplexity. This is intuitive since larger number of codebooks leads to better representation of the values in each block since LO-BCQ allows it to map to the codebook

Table 3: (a) PTQ Perplexity (lower is better) on Wikitext-103 dataset and MMLU accuracy (higher is better) with Nemotron4-15B model, and (b) Comparing perplexity loss of weight-only (W4A8) LO-BCQ to other weight-only quantization works such as GPTQ, AWQ, OmniQ and QuiP#. Here, the LO-BCQ configuration is denoted by tuple $\{L_A, L_b, N_c\}$ = {Length of block array, Length of block, Number of codebooks}.

| Method | Bitwidth | Wiki3 PPL ($\Delta$) | MMLU (5-shot) Avg ($\Delta$ %) |
|---|---|---|---|
| Nemotron4-15B | | | |
| FP32 | 32 | 5.87 | 64.3 |
| MX4 | 4.5 | 8.88 (3.01) | 58.15 (6.15) |
| VSQ | 4.5 | 7.58 (1.71) | 57.38 (6.92) |
| MXFP4 | 4.25 | 8.24 (2.37) | 58.28 (6.02) |
| **LO-BCQ** $\{64, 8, 2\}$ | 4.25 | 6.30 (0.43) | 63.17 (1.13) |
| **LO-BCQ** $\{64, 8, 8\}$ | 4.5 | 6.13 (0.26) | **63.72 (0.58)** |
| **LO-BCQ** $\{32, 8, 16\}$ | 4.75 | 6.03 (0.16) | **64.33 (+0.03)** |

(a)

| Method | Llama2-7B | Llama2-70B |
|---|---|---|
| GPTQ | 0.22 | 0.10 |
| AWQ | 0.13 | **0.09** |
| OmniQ | 0.27 | 0.15 |
| QuiP# | 0.19 | 0.10 |
| **LO-BCQ** $\{128, 8, 2\}$ | 0.14 | **0.09** |
| **LO-BCQ** $\{128, 8, 4\}$ | 0.12 | **0.07** |
| **LO-BCQ** $\{128, 8, 8\}$ | **0.09** | **0.06** |
| **LO-BCQ** $\{128, 8, 16\}$ | **0.08** | **0.05** |

(b)

Table 4: Ablation studies: (a) Perplexity on Wikitext-103 dataset across various LO-BCQ configurations, and (b) Perplexity on Wikitext-103 dataset with universally calibrated vs locally calibrated codebooks

| $L_b \rightarrow$ | 8 | | | | 4 | | 2 |
|---|---|---|---|---|---|---|---|
| $N_c$ / $L_A$ | 2 | 4 | 8 | 16 | 2 | 4 | 2 |
| Llama2-70B (FP32 PPL = 3.14) | | | | | | | |
| 64 | 3.35 | 3.25 | 3.23 | 3.21 | 3.31 | 3.22 | 3.27 |
| 32 | 3.27 | 3.24 | 3.22 | 3.20 | 3.25 | 3.22 | 3.22 |
| 16 | 3.25 | 3.22 | 3.20 | 3.19 | 3.23 | 3.20 | 3.20 |
| GPT3-22B (FP32 PPL = 6.54) | | | | | | | |
| 64 | 6.74 | 6.64 | 6.62 | 6.63 | 6.71 | 6.64 | 6.64 |
| 32 | 6.67 | 6.64 | 6.61 | 6.59 | 6.65 | 6.64 | 6.60 |
| 16 | 6.67 | 6.63 | 6.59 | 6.61 | 6.66 | 6.63 | 6.62 |

(a)

| Llama2-7B (FP32 PPL = 5.06), $L_b = 8$ | | | | |
|---|---|---|---|---|
| $N_c$ / $L_A$ | 2 | 4 | 8 | 16 |
| Universally Calibrated Codebooks | | | | |
| 64 | 5.31 | 5.26 | 5.19 | 5.18 |
| 32 | 5.23 | 5.25 | 5.18 | 5.15 |
| 16 | 5.23 | 5.19 | 5.16 | 5.14 |
| Layerwise Calibrated Codebooks | | | | |
| 64 | 5.29 | 5.22 | 5.19 | 5.17 |
| 32 | 5.23 | 5.19 | 5.17 | 5.15 |
| 16 | 5.20 | 5.17 | 5.15 | 5.14 |

(b)

with best representation. Further, when the block array size is reduced, we achieve better perplexity. The block array corresponds to the granularity of normalization. As discussed in section 2.3, normalization improves convergence of LO-BCQ and results in better perplexity. Further, when comparing configurations with same bitwidth (see Table 1), we find that the configuration with larger number of codebooks is better than smaller block array. This shows that the per-block metadata is better utilized for codebook selectors than scale factors.

Furthermore, we find that reducing the block length ($L_b$) below 8 results in diminishing returns. This is because, the overhead of storing codebook selectors is larger for a smaller block. For a given bitwidth, configuration with smaller $L_b$ has fewer codebooks. Therefore, these configurations result in larger loss in perplexity.

Table 4b compares the perplexity with universally calibrated codebooks to codebooks calibrated layerwise (per-tensor) in Llama2-7B model. The layerwise calibrated codebooks achieve slightly better perplexity when the number of codebooks are small (e.g. $N_c = 2$). However, they do not provide significant benefits when $N_c > 4$ despite the comparatively larger calibration effort. Therefore, in our experiments in this paper, we have largely explored universally calibrated codebooks.

## 5 CONCLUSION

The inference accuracy of LLMs during per-block (fine-grained) quantization is significantly influenced by the number format of the operands and per-block scale factors. Several previous works have explored novel number formats to improve accuracy. However, none have explored per-block quantization methods involving clustering that minimize quantization MSE. In this work, we propose LO-BCQ, an iterative block clustering and quantization algorithm that greedily minimizes quantization MSE for any operand (weights and activations) through locally optimal steps at each step of the iteration. We demonstrate that LO-BCQ achieves state-of-the-art perplexity across a suite of GPT3, LLama2 and Nemotron4 models on various downstream tasks such Wikitext-103, LM evaluation harness and MMLU.

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

# A APPENDIX

## A.1 LLOYD-MAX ALGORITHM

For a given quantization bitwidth $B$ and an operand $\boldsymbol{X}$, the Lloyd-Max algorithm finds $2^B$ quantization levels $\{\hat{x}_i\}_{i=1}^{2^B}$ such that quantizing $\boldsymbol{X}$ by rounding each scalar in $\boldsymbol{X}$ to the nearest quantization level minimizes the quantization MSE.

The algorithm starts with an initial guess of quantization levels and then iteratively computes quantization thresholds $\{\tau_i\}_{i=1}^{2^B-1}$ and updates quantization levels $\{\hat{x}_i\}_{i=1}^{2^B}$. Specifically, at iteration $n$, thresholds are set to the midpoints of the previous iteration's levels:

$$\tau_i^{(n)} = \frac{\hat{x}_i^{(n-1)} + \hat{x}_{i+1}^{(n-1)}}{2} \text{ for } i = 1 \dots 2^B - 1$$

Subsequently, the quantization levels are re-computed as conditional means of the data regions defined by the new thresholds:

$$\hat{x}_i^{(n)} = \mathbb{E}\left[\boldsymbol{X} \big| \boldsymbol{X} \in [\tau_{i-1}^{(n)}, \tau_i^{(n)}]\right] \text{ for } i = 1 \dots 2^B$$

where to satisfy boundary conditions we have $\tau_0 = -\infty$ and $\tau_{2^B} = \infty$. The algorithm iterates the above steps until convergence.

Figure 8: Quantization levels and the corresponding quantization MSE of Floating Point (left) vs Lloyd-Max (right) Quantizers for a layer of weights in the GPT3-126M model.

Table 5: Comparing perplexity (lower is better) achieved by floating point quantizers and Lloyd-Max quantizers on a GPT3-126M model for the Wikitext-103 dataset.

| Bitwidth | Floating-Point Quantizer | | Lloyd-Max Quantizer |
| --- | --- | --- | --- |
| | Best Format | Perplexity | Perplexity |
| 7 | E3M3 | 18.32 | 18.27 |
| 6 | E3M2 | 19.07 | 18.51 |
| 5 | E4M0 | 43.89 | 19.71 |

Figure 8 compares the quantization levels of a 7-bit floating point (E3M3) quantizer (left) to a 7-bit Lloyd-Max quantizer (right) when quantizing a layer of weights from the GPT3-126M model at a per-tensor granularity. As shown, the Lloyd-Max quantizer achieves substantially lower quantization MSE. Further, Table 5 shows the superior perplexity achieved by Lloyd-Max quantizers for bitwidths of 7, 6 and 5. The difference between the quantizers is clear at 5 bits, where per-tensor FP quantization incurs a drastic and unacceptable increase in perplexity, while Lloyd-Max quantization incurs a much smaller increase. Nevertheless, we note that even the optimal Lloyd-Max quantizer incurs a notable ($\sim 1.5$) increase in perplexity due to the coarse granularity of quantization.

### A.2 PROOF OF LOCAL OPTIMALITY OF LO-BCQ

For a given block $\boldsymbol{b}_j$, the quantization MSE during LO-BCQ can be empirically evaluated as $\frac{1}{L_b}\|\boldsymbol{b}_j - \hat{\boldsymbol{b}}_j\|_2^2$ where $\hat{\boldsymbol{b}}_j$ is computed from equation (1) as $C_{f(\boldsymbol{b}_j)}(\boldsymbol{b}_j)$. Further, for a given block cluster $\mathcal{B}_i$, we compute the quantization MSE as $\frac{1}{|\mathcal{B}_i|}\sum_{\boldsymbol{b}\in\mathcal{B}_i}\frac{1}{L_b}\|\boldsymbol{b} - C_i^{(n)}(\boldsymbol{b})\|_2^2$. Therefore, at the end of iteration $n$, we evaluate the overall quantization MSE $J^{(n)}$ for a given operand $\boldsymbol{X}$ composed of $N_c$ block clusters as:

$$J^{(n)} = \frac{1}{N_c}\sum_{i=1}^{N_c}\frac{1}{|\mathcal{B}_i^{(n)}|}\sum_{\boldsymbol{v}\in\mathcal{B}_i^{(n)}}\frac{1}{L_b}\|\boldsymbol{b} - B_i^{(n)}(\boldsymbol{b})\|_2^2$$

At the end of iteration $n$, the codebooks are updated from $\mathcal{C}^{(n-1)}$ to $\mathcal{C}^{(n)}$. However, the mapping of a given vector $\boldsymbol{b}_j$ to quantizers $\mathcal{C}^{(n)}$ remains as $f^{(n)}(\boldsymbol{b}_j)$. At the next iteration, during the vector clustering step, $f^{(n+1)}(\boldsymbol{b}_j)$ finds new mapping of $\boldsymbol{b}_j$ to updated codebooks $\mathcal{C}^{(n)}$ such that the quantization MSE over the candidate codebooks is minimized. Therefore, we obtain the following result for $\boldsymbol{b}_j$:

$$\frac{1}{L_b}\|\boldsymbol{b}_j - C_{f^{(n+1)}(\boldsymbol{b}_j)}^{(n)}(\boldsymbol{b}_j)\|_2^2 \leq \frac{1}{L_b}\|\boldsymbol{b}_j - C_{f^{(n)}(\boldsymbol{b}_j)}^{(n)}(\boldsymbol{b}_j)\|_2^2$$

That is, quantizing $\boldsymbol{b}_j$ at the end of the block clustering step of iteration $n + 1$ results in lower quantization MSE compared to quantizing at the end of iteration $n$. Since this is true for all $\boldsymbol{b} \in \boldsymbol{X}$, we assert the following:

$$\tilde{J}^{(n+1)} = \frac{1}{N_c}\sum_{i=1}^{N_c}\frac{1}{|\mathcal{B}_i^{(n+1)}|}\sum_{\boldsymbol{b}\in\mathcal{B}_i^{(n+1)}}\frac{1}{L_b}\|\boldsymbol{b} - C_i^{(n)}(b)\|_2^2 \leq J^{(n)} \tag{9}$$

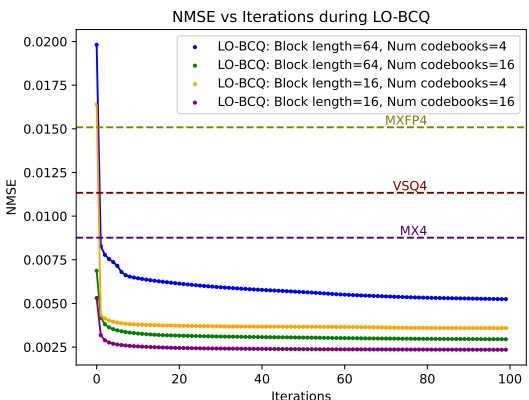

Figure 9: NMSE vs interations during LO-BCQ compared to other block quantization proposals

where $\tilde{J}^{(n+1)}$ is the the quantization MSE after the vector clustering step at iteration $n+1$.

Next, during the codebook update step (6) at iteration $n+1$, the per-cluster codebooks $\mathcal{C}^{(n)}$ are updated to $\mathcal{C}^{(n+1)}$ by invoking the Lloyd-Max algorithm (Lloyd, 1982). We know that for any given value distribution, the Lloyd-Max algorithm minimizes the quantization MSE. Therefore, for a given vector cluster $\mathcal{B}_i$ we obtain the following result:

$$\frac{1}{|\mathcal{B}_i^{(n+1)}|} \sum_{\boldsymbol{b} \in \mathcal{B}_i^{(n+1)}} \frac{1}{L_b} \|\boldsymbol{b} - C_i^{(n+1)}(\boldsymbol{b})\|_2^2 \quad \leq \quad \frac{1}{|\mathcal{B}_i^{(n+1)}|} \sum_{\boldsymbol{b} \in \mathcal{B}_i^{(n+1)}} \frac{1}{L_b} \|\boldsymbol{b} - C_i^{(n)}(\boldsymbol{b})\|_2^2 \quad (10)$$

The above equation states that quantizing the given block cluster $\mathcal{B}_i$ after updating the associated codebook from $C_i^{(n)}$ to $C_i^{(n+1)}$ results in lower quantization MSE. Since this is true for all the block clusters, we derive the following result:

$$J^{(n+1)} = \frac{1}{N_c} \sum_{i=1}^{N_c} \frac{1}{|\mathcal{B}_i^{(n+1)}|} \sum_{\boldsymbol{b} \in \mathcal{B}_i^{(n+1)}} \frac{1}{L_b} \|\boldsymbol{b} - C_i^{(n+1)}(\boldsymbol{b})\|_2^2 \leq \tilde{J}^{(n+1)} \tag{11}$$

Following (9) and (11), we find that the quantization MSE is non-increasing for each iteration, that is, $J^{(1)} \geq J^{(2)} \geq J^{(3)} \geq \ldots \geq J^{(M)}$ where $M$ is the maximum number of iterations. ∎

Figure 9 shows the empirical convergence of LO-BCQ across several block lengths and number of codebooks. Also, the MSE achieved by LO-BCQ is compared to baselines such as MXFP and VSQ. As shown, LO-BCQ converges to a lower MSE than the baselines. Further, we achieve better convergence for larger number of codebooks ($N_c$) and for a smaller block length ($L_b$), both of which increase the bitwidth of BCQ (see Eq 3).

### A.3 Additional Accuracy Results

### A.4 Number Formats and Quantization Method

#### A.4.1 Integer Format

An $n$-bit signed integer (INT) is typically represented with a 2s-complement format (Yao et al., 2022; Xiao et al., 2023; Dai et al., 2021), where the most significant bit denotes the sign.

#### A.4.2 Floating Point Format

An $n$-bit signed floating point (FP) number $x$ comprises of a 1-bit sign ($x_{\text{sign}}$), $B_m$-bit mantissa ($x_{\text{mant}}$) and $B_e$-bit exponent ($x_{\text{exp}}$) such that $B_m + B_e = n - 1$. The associated constant exponent bias ($E_{\text{bias}}$) is computed as ($2^{B_e-1} - 1$). We denote this format as $E_{B_e} M_{B_m}$.

| $L_b \rightarrow$ | 8 | | | | 4 | | 2 |
|---|---|---|---|---|---|---|---|
| $N_c$ / $L_A$ | 2 | 4 | 8 | 16 | 2 | 4 | 2 |
| GPT3-1.3B (FP32 PPL = 9.98) | | | | | | | |
| 64 | 10.40 | 10.23 | 10.17 | 10.15 | 10.28 | 10.18 | 10.19 |
| 32 | 10.25 | 10.20 | 10.15 | 10.12 | 10.23 | 10.17 | 10.17 |
| 16 | 10.22 | 10.16 | 10.10 | 10.09 | 10.21 | 10.14 | 10.16 |
| GPT3-8B (FP32 PPL = 7.38) | | | | | | | |
| 64 | 7.61 | 7.52 | 7.48 | 7.47 | 7.55 | 7.49 | 7.50 |
| 32 | 7.52 | 7.50 | 7.46 | 7.45 | 7.52 | 7.48 | 7.48 |
| 16 | 7.51 | 7.48 | 7.44 | 7.44 | 7.51 | 7.49 | 7.47 |

Table 6: Wikitext-103 perplexity across GPT3-1.3B and 8B models.

| $L_b \rightarrow$ | 8 | | | |
|---|---|---|---|---|
| $N_c$ / $L_A$ | 2 | 4 | 8 | 16 |
| Llama2-7B (FP32 PPL = 5.06) | | | | |
| 64 | 5.31 | 5.26 | 5.19 | 5.18 |
| 32 | 5.23 | 5.25 | 5.18 | 5.15 |
| 16 | 5.23 | 5.19 | 5.16 | 5.14 |
| Nemotron4-15B (FP32 PPL = 5.87) | | | | |
| 64 | 6.3 | 6.20 | 6.13 | 6.08 |
| 32 | 6.24 | 6.12 | 6.07 | 6.03 |
| 16 | 6.12 | 6.14 | 6.04 | 6.02 |
| Nemotron4-340B (FP32 PPL = 3.48) | | | | |
| 64 | 3.67 | 3.62 | 3.60 | 3.59 |
| 32 | 3.63 | 3.61 | 3.59 | 3.56 |
| 16 | 3.61 | 3.58 | 3.57 | 3.55 |

Table 7: Wikitext-103 perplexity compared to FP32 baseline in Llama2-7B and Nemotron4-15B, 340B models

| $L_b \rightarrow$ | 8 | | | | 8 | | | |
|---|---|---|---|---|---|---|---|---|
| $N_c$ / $L_A$ | 2 | 4 | 8 | 16 | 2 | 4 | 8 | 16 |
| Llama2-7B (FP32 Accuracy = 45.8%) | | | | Llama2-70B (FP32 Accuracy = 69.12%) | | | | |
| 64 | 43.9 | 43.4 | 43.9 | 44.9 | 68.07 | 68.27 | 68.17 | 68.75 |
| 32 | 44.5 | 43.8 | 44.9 | 44.5 | 68.37 | 68.51 | 68.35 | 68.27 |
| 16 | 43.9 | 42.7 | 44.9 | 45 | 68.12 | 68.77 | 68.31 | 68.59 |
| GPT3-22B (FP32 Accuracy = 38.75%) | | | | Nemotron4-15B (FP32 Accuracy = 64.3%) | | | | |
| 64 | 36.71 | 38.85 | 38.13 | 38.92 | 63.17 | 62.36 | 63.72 | 64.09 |
| 32 | 37.95 | 38.69 | 39.45 | 38.34 | 64.05 | 62.30 | 63.8 | 64.33 |
| 16 | 38.88 | 38.80 | 38.31 | 38.92 | 63.22 | 63.51 | 63.93 | 64.43 |

Table 8: Accuracy on MMLU dataset across GPT3-22B, Llama2-7B, 70B and Nemotron4-15B models.

| $L_b \to$ | 8 | | | | 8 | | | |
|---|---|---|---|---|---|---|---|---|
| $N_c$ / $L_A$ | 2 | 4 | 8 | 16 | 2 | 4 | 8 | 16 |
| Race (FP32 Accuracy = 37.51%) | | | | | Boolq (FP32 Accuracy = 64.62%) | | | |
| 64 | 36.94 | 37.13 | 36.27 | 37.13 | 63.73 | 62.26 | 63.49 | 63.36 |
| 32 | 37.03 | 36.36 | 36.08 | 37.03 | 62.54 | 63.51 | 63.49 | 63.55 |
| 16 | 37.03 | 37.03 | 36.46 | 37.03 | 61.1 | 63.79 | 63.58 | 63.33 |
| Winogrande (FP32 Accuracy = 58.01%) | | | | | Piqa (FP32 Accuracy = 74.21%) | | | |
| 64 | 58.17 | 57.22 | 57.85 | 58.33 | 73.01 | 73.07 | 73.07 | 72.80 |
| 32 | 59.12 | 58.09 | 57.85 | 58.41 | 73.01 | 73.94 | 72.74 | 73.18 |
| 16 | 57.93 | 58.88 | 57.93 | 58.56 | 73.94 | 72.80 | 73.01 | 73.94 |

Table 9: Accuracy on LM evaluation harness tasks on GPT3-1.3B model.

| $L_b \to$ | 8 | | | | 8 | | | |
|---|---|---|---|---|---|---|---|---|
| $N_c$ / $L_A$ | 2 | 4 | 8 | 16 | 2 | 4 | 8 | 16 |
| Race (FP32 Accuracy = 41.34%) | | | | | Boolq (FP32 Accuracy = 68.32%) | | | |
| 64 | 40.48 | 40.10 | 39.43 | 39.90 | 69.20 | 68.41 | 69.45 | 68.56 |
| 32 | 39.52 | 39.52 | 40.77 | 39.62 | 68.32 | 67.43 | 68.17 | 69.30 |
| 16 | 39.81 | 39.71 | 39.90 | 40.38 | 68.10 | 66.33 | 69.51 | 69.42 |
| Winogrande (FP32 Accuracy = 67.88%) | | | | | Piqa (FP32 Accuracy = 78.78%) | | | |
| 64 | 66.85 | 66.61 | 67.72 | 67.88 | 77.31 | 77.42 | 77.75 | 77.64 |
| 32 | 67.25 | 67.72 | 67.72 | 67.00 | 77.31 | 77.04 | 77.80 | 77.37 |
| 16 | 68.11 | 68.90 | 67.88 | 67.48 | 77.37 | 78.13 | 78.13 | 77.69 |

Table 10: Accuracy on LM evaluation harness tasks on GPT3-8B model.

| $L_b \to$ | 8 | | | | 8 | | | |
|---|---|---|---|---|---|---|---|---|
| $N_c$ / $L_A$ | 2 | 4 | 8 | 16 | 2 | 4 | 8 | 16 |
| Race (FP32 Accuracy = 40.67%) | | | | | Boolq (FP32 Accuracy = 76.54%) | | | |
| 64 | 40.48 | 40.10 | 39.43 | 39.90 | 75.41 | 75.11 | 77.09 | 75.66 |
| 32 | 39.52 | 39.52 | 40.77 | 39.62 | 76.02 | 76.02 | 75.96 | 75.35 |
| 16 | 39.81 | 39.71 | 39.90 | 40.38 | 75.05 | 73.82 | 75.72 | 76.09 |
| Winogrande (FP32 Accuracy = 70.64%) | | | | | Piqa (FP32 Accuracy = 79.16%) | | | |
| 64 | 69.14 | 70.17 | 70.17 | 70.56 | 78.24 | 79.00 | 78.62 | 78.73 |
| 32 | 70.96 | 69.69 | 71.27 | 69.30 | 78.56 | 79.49 | 79.16 | 78.89 |
| 16 | 71.03 | 69.53 | 69.69 | 70.40 | 78.13 | 79.16 | 79.00 | 79.00 |

Table 11: Accuracy on LM evaluation harness tasks on GPT3-22B model.

| $L_b \to$ | 8 | | | | 8 | | | |
|---|---|---|---|---|---|---|---|---|
| $N_c$ / $L_A$ | 2 | 4 | 8 | 16 | 2 | 4 | 8 | 16 |
| Race (FP32 Accuracy = 44.4%) | | | | | Boolq (FP32 Accuracy = 79.29%) | | | |
| 64 | 42.49 | 42.51 | 42.58 | 43.45 | 77.58 | 77.37 | 77.43 | 78.1 |
| 32 | 43.35 | 42.49 | 43.64 | 43.73 | 77.86 | 75.32 | 77.28 | 77.86 |
| 16 | 44.21 | 44.21 | 43.64 | 42.97 | 78.65 | 77 | 76.94 | 77.98 |
| Winogrande (FP32 Accuracy = 69.38%) | | | | | Piqa (FP32 Accuracy = 78.07%) | | | |
| 64 | 68.9 | 68.43 | 69.77 | 68.19 | 77.09 | 76.82 | 77.09 | 77.86 |
| 32 | 69.38 | 68.51 | 68.82 | 68.90 | 78.07 | 76.71 | 78.07 | 77.86 |
| 16 | 69.53 | 67.09 | 69.38 | 68.90 | 77.37 | 77.8 | 77.91 | 77.69 |

Table 12: Accuracy on LM evaluation harness tasks on Llama2-7B model.

| $L_b \rightarrow$ | 8 | | | | 8 | | | |
|---|---|---|---|---|---|---|---|---|
| $N_c$ / $L_A$ | 2 | 4 | 8 | 16 | 2 | 4 | 8 | 16 |
| Race (FP32 Accuracy = 48.8%) | | | | | Boolq (FP32 Accuracy = 85.23%) | | | |
| 64 | 49.00 | 49.00 | 49.28 | 48.71 | 82.82 | 84.28 | 84.03 | 84.25 |
| 32 | 49.57 | 48.52 | 48.33 | 49.28 | 83.85 | 84.46 | 84.31 | 84.93 |
| 16 | 49.85 | 49.09 | 49.28 | 48.99 | 85.11 | 84.46 | 84.61 | 83.94 |
| Winogrande (FP32 Accuracy = 79.95%) | | | | | Piqa (FP32 Accuracy = 81.56%) | | | |
| 64 | 78.77 | 78.45 | 78.37 | 79.16 | 81.45 | 80.69 | 81.45 | 81.5 |
| 32 | 78.45 | 79.01 | 78.69 | 80.66 | 81.56 | 80.58 | 81.18 | 81.34 |
| 16 | 79.95 | 79.56 | 79.79 | 79.72 | 81.28 | 81.66 | 81.28 | 80.96 |

Table 13: Accuracy on LM evaluation harness tasks on Llama2-70B model.

### A.4.3  MX FORMAT

The MX format proposed in (Rouhani et al., 2023a) introduces the concept of sub-block shifting. For every two scalar elements of $b$-bits each, there is a shared exponent bit. The value of this exponent bit is determined through an empirical analysis that targets minimizing quantization MSE. We note that the FP format $E_1 M_b$ is strictly better than MX from an accuracy perspective since it allocates a dedicated exponent bit to each scalar as opposed to sharing it across two scalars. Therefore, we conservatively bound the accuracy of a $b + 2$-bit signed MX format with that of a $E_1 M_b$ format in our comparisons. For instance, we use E1M2 format as a proxy for MX4.

### A.4.4  QUANTIZATION SCHEME

A quantization scheme dictates how a given unquantized tensor is converted to its quantized representation. We consider FP formats for the purpose of illustration. Given an unquantized tensor $\boldsymbol{X}$ and an FP format $E_{B_e} M_{B_m}$, we first, we compute the quantization scale factor $s_X$ that maps the maximum absolute value of $\boldsymbol{X}$ to the maximum quantization level of the $E_{B_e} M_{B_m}$ format as follows:

$$s_X = \frac{\max(|\boldsymbol{X}|)}{\max(E_{B_e} M_{B_m})} \tag{12}$$

In the above equation, $|\cdot|$ denotes the absolute value function.

Next, we scale $\boldsymbol{X}$ by $s_X$ and quantize it to $\hat{\boldsymbol{X}}$ by rounding it to the nearest quantization level of $E_{B_e} M_{B_m}$ as:

$$\hat{\boldsymbol{X}} = \text{round-to-nearest}\left(\frac{\boldsymbol{X}}{s_X}, E_{B_e} M_{B_m}\right) \tag{13}$$

We perform dynamic max-scaled quantization (Wu et al., 2020b), where the scale factor $s$ for activations is dynamically computed during runtime.

### A.5  VECTOR SCALED QUANTIZATION

During VSQ (Dai et al., 2021), the operand tensors are decomposed into 1D vectors in a hardware friendly manner as shown in Figure 10. Since the decomposed tensors are used as operands in matrix multiplications during inference, it is beneficial to perform this decomposition along the reduction dimension of the multiplication. The vectorwise quantization is performed similar to tensorwise quantization described in Equations 12 and 13, where a scale factor $s_v$ is required for each vector $\boldsymbol{v}$ that maps the maximum absolute value of that vector to the maximum quantization level. While smaller vector lengths can lead to larger accuracy gains, the associated

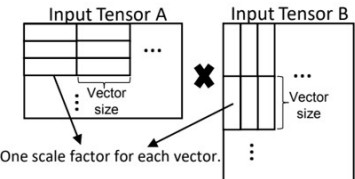

Figure 10: Vectorwise decomposition for per-vector scaled quantization (VSQ (Dai et al., 2021)).

memory and computational overheads due to the per-vector scale factors increases. To alleviate these overheads, VSQ (Dai et al., 2021) proposed a second level quantization of the per-vector scale factors to unsigned integers, while MX (Rouhani et al., 2023b) quantizes them to integer powers of 2 (denoted as $2^{INT}$).

