# OpenReview forum: "BCQ: Block Clustered Quantization for 4-bit (W4A4) LLM inference"
_ICLR.cc/2025/Conference — Submitted to ICLR 2025_

### Official Review · Reviewer_ZF8x · 2024-10-28

**Soundness:** 2
**Presentation:** 2
**Contribution:** 3
**Rating:** 6
**Confidence:** 3

**Summary:**

This paper introduce a novel block quantization method called LO-BCQ, which achieves state-of-the-art level while avoiding additional memory overheads by iterative block clustering and quantization through locally optimization pipeline.

**Strengths:**

1. This paper iteratively optimizes block clustering and per-cluster quantization to minimize MSE loss for both weight and activation and achieves advanced level.
2. LO-BCQ avoids memory overheads caused by per-block specific codebook and scaling factors in previous methods by sharing scaling factors in a group of blocks.

**Weaknesses:**

1. The paper ignores to describe the distinctions between block quantization and group-wise quantization. because from the perspective of form and definition, there seems to be no difference between them.
2. This paper solely conduct comparison experiments with block quantization based methods. However, there are many other widely used PTQ methods for LLMs such as OmniQuant, GPTQ, QuIP, etc.
3. In addition to accuracy, efficiency is an important criterion for measuring the effectiveness of a proposed method. Compared to general PTQ methods, LO-BCQ involves numerous optimization processes, which to some extent reduce the method's efficiency.
4. The logical flow of the writing needs to be improved, especially for Introduction.

**Questions:**

1. For weakness1, if feasible, please involve detailed analysis and experiments for block quantization and group-wise quantization, such as AWQ-g128.
2. For weakness2, to demonstrate the advanced nature of your method, please include comparisons with the mentioned PTQ methods in your experimental tables.
3. For weakness3, Please provide the running time consumption of LO-BCQ. If the proposed method is faster than OmniQuant, which is currently known for a relatively low-efficient but effective PTQ method, it would still be within an acceptable range.
4. The author should check for some spelling errors in the manuscript, such as "biwidth" in line 054.

---

> ### Author Response · Authors · 2024-11-27
>
> We greatly appreciate the comments and questions of the reviewer. We thank the reviewer for acknowledging our locally optimal block clustering approach for quantization that achieves current state-of-the-art during 4-bit quantization of both weights and activations across several LLMs and datasets.
>
> The reviewer has raised valid criticism regarding the terminology of block quantization vs group quantization. First, we would like to apologize for this confusion and clarify that the group quantization and block quantization methods are the same. As quantization for LLMs is a rapidly evolving field, there hasn’t been a consensus on unified terminology for this approach. Further, we would like to emphasize that LO-BCQ improves over group-quantization that typically quantizes each group to a given number format by allowing each group to map to a set of codebooks that minimize quantization error.
>
> To demonstrate the advanced nature of our method, we perform detailed comparison against the weight-only quantization proposals listed by the reviewer. While the techniques in Omniquant, GPTQ, AWQ and QuiP have been demonstrated in weight quantization, we would like to highlight that LO-BCQ can be applied to both weights and activations. That is the codebooks we calibrate through LO-BCQ are universal across operand types and models. Further, while LO-BCQ is a post-training quantization (PTQ) method, GPTQ and QuiP involves finetuning to recover accuracy loss from quantization. We have compared the Wikitext perplexity loss of LO-BCQ against these works in Figure 1 of our paper.
>
> In the table below, we compare weight-only LO-BCQ to AWQ and GPTQ, all with group size = 128 for a fair comparison. As shown, LO-BCQ achieves comparatively lower loss in perplexity. Further, we evaluate this loss on Wikitext-103 dataset, which is a much larger dataset compared to Wikitext2 which is used by GPTQ and AWQ.
>
> # Wikitext PPL Loss Compared to Unquantized Baseline
>
> | Model        | GPTQ | AWQ  | LO-BCQ (2 Codebooks) | LO-BCQ (4 Codebooks) | LO-BCQ (8 Codebooks) |
> |--------------|------|------|----------------------|----------------------|----------------------|
> | LLAMA2-7B    | 0.22 | 0.13 | 0.14                 | 0.12                 | 0.09                 |
> | LLAMA2-70B   | 0.10 | 0.09 | 0.09                 | 0.07                 | 0.06                 |
>
> Additionally, in the table below, we compare against Omniquant and QuiP across various datasets such as Wikitext, PIQA and Winogrande. As shown, LO-BCQ achieves significant perplexity improvement over OmniQuant and QuiP. Further, the loss in accuracy in PIQA and Winogrande datasets are maintained to be <1% with 8 codebooks.
>
> # Wikitext PPL Loss Compared to Unquantized Baseline
>
> | Model        | Omniquant | QuiP | LO-BCQ (2 Codebooks) | LO-BCQ (4 Codebooks) | LO-BCQ (8 Codebooks) |
> |--------------|-----------|------|----------------------|----------------------|----------------------|
> | LLAMA2-7B    | 0.27      | 0.19 | 0.14                 | 0.12                 | 0.09                 |
> | LLAMA2-70B   | 0.15      | 0.10 | 0.09                 | 0.07                 | 0.06                 |
>
> # PIQA Accuracy
>
> | Model        | Unquantized Baseline | Omniquant | QuiP | LO-BCQ (2 Codebooks) | LO-BCQ (4 Codebooks) | LO-BCQ (8 Codebooks) |
> |--------------|----------------------|-----------|------|----------------------|----------------------|----------------------|
> | LLAMA2-7B    | 78.07                | 77.1      | 78.4 | 77.97                | 77.97                | 77.58                |
> | LLAMA2-70B   | 81.56                | 80.7      | 81.4 | 81.18                | 81.34                | 81.45                |
>
> # Winogrande Accuracy
>
> | Model        | Unquantized Baseline | Omniquant | QuiP | LO-BCQ (2 Codebooks) | LO-BCQ (4 Codebooks) | LO-BCQ (8 Codebooks) |
> |--------------|----------------------|-----------|------|----------------------|----------------------|----------------------|
> | LLAMA2-7B    | 69.38                | 67.0      | 67.6 | 68.27                | 69.30                | 69.69                |
> | LLAMA2-70B   | 79.95                | 75.8      | 77.1 | 79.32                | 79.79                | 80.03                |

---

> > ### Author Response · Authors · 2024-11-27
> >
> > We agree with the reviewer that latency of inference is an important metric to evaluate the performance of our proposal. For this rebuttal, given the limited time, we would like to qualitatively discuss the performance of LO-BCQ. As demonstrated in Keller et al [3], with a dedicated inference accelerator for 64-wide group quantization, considerable speedup can be achieved, especially during the pre-fill phase in LLMs involving GEMM computations. In LO-BCQ, in addition to group quantization, we perform a decoding step where each 4-bit scalar is decoded to 6-bit integers. As a result of a small number of codebooks utilized by LO-BCQ (footprint = 0.19KB), one can potentially store them in shared memory of each SM resulting in a small look-up cost. A dedicated hardware design can amortize the cost of this lookup through spatial and temporal re-use of operands. We would like to emphasize that, as a result of a small number of codebooks (<= 16) in our work, the cost of this lookup is considerably small compared to other codebook-based proposals such as QuIP[4] and AQLM[5] where the number of codebooks are as large as 2^16. We would like to clarify that the clustering and calibration of codebooks is performed prior to inference (offline). We freeze the codebooks identified by LO-BCQ algorithm across our evaluations.

---

### Official Review · Reviewer_oJwf · 2024-10-30

**Soundness:** 2
**Presentation:** 3
**Contribution:** 2
**Rating:** 5
**Confidence:** 5

**Summary:**

This paper introduces a method named block clustered quantization (BCQ), which decomposes operand tensors into blocks, clustering them based on statistics and designing codebooks for each cluster. When weight and activation scalars are encoded to W4A4 format, BCQ achieves the current state-of-the-art by demonstrating < 1% loss in inference accuracy across several LLMs.

**Strengths:**

1. This paper is well written, clearly described, and presents an effective methodology that makes group quantization a significant enhancement across a wide range of tasks and models;
2. Combining a quantization problem with a clustering problem is novel enough, I think, and the author's algorithm is simple and effective enough.

**Weaknesses:**

1. Although the authors' group quantization achieved a significant improvement in accuracy, the paper does not report results on inference speed. From QServe we can learn that group quantization is very inefficient in terms of speed of reasoning. Does the authors' proposed quantization scheme have any performance advantage over group quantization, except for the advantage of saving some storage? Can the authors provide the corresponding latency metrics?
2. The authors' experiments should include other model types such as LLaMA3 and Mistral, or even LLaMA3.1 and LLaMA3.2, and I am very much looking forward to the performance of the authors' method on MOE LLM model.
3. Figure 2, Figure 3 and Figure 4 can be modified to better illustrate and/or an algorithm description of the steps.

**Questions:**

1. Has the author's algorithm been compared to GPTQ, which is a very effective way to quantify large models in PTQ quantization, and how does the author's method compare to GPTQ in terms of effectiveness and efficiency?

---

> ### Comment · Reviewer_oJwf · 2024-11-27
>
> Since there is no feedback, I keep my rating to reject the work.

---

> > ### Author Response · Authors · 2024-11-27
> >
> > We greatly appreciate the comments and questions of the reviewer. We thank the reviewer for acknowledging our novel and effective clustering approach for quantization that achieves current state-of-the-art by demonstrating <1% loss in inference accuracy across several LLMs and datasets during 4-bit quantization of both weights and activations.
> >
> > The reviewer’s main concern is regarding the cost of group quantization and how LO-BCQ can improve over group quantization. We agree with the reviewer that group quantization is computationally expensive compared to conventional quantization methods due to the scale factor computations per-group -- the smaller the group-size, the larger the cost of this computation. However, for a given group-size, LO-BCQ achieves significantly better inference accuracy compared to other group quantization methods since LO-BCQ can flexibly choose a codebook from a set of codebooks. As a result, LO-BCQ enables increasing group size by minimally trading-off accuracy, reducing both storage and computational costs. We demonstrate this behavior in the table below, where LO-BCQ maintains the perplexity loss on Wikitext-103 dataset to <0.1 despite the large group size of 64 compared to other works such as MXFP4[1] and MX4[2].
> >
> > # LLAMA2-70B: Loss Compared to Unquantized Baseline (Lower is Better)
> >
> > | Method            | Group Size | Wiki-103 | PIQA | BoolQ | HellaSwag | Winogrande |
> > |-------------------|------------|----------|------|-------|-----------|------------|
> > | LO-BCQ (8 CODEBOOKS)  | 64         | 0.09     | 0.11 | 1.20  | 0.51      | 1.58       |
> > | LO-BCQ (16 CODEBOOKS) | 64         | 0.07     | 0.06 | 0.98  | 0.23      | 0.79       |
> > | MXFP4[1]              | 32         | 0.54     | 0.98 | 2.17  | 2.03      | 3.63       |
> > | MX4[2]                | 16         | 0.44     | 0.98 | 2.82  | 2.03      | 3.55       |
> >
> > We agree with the reviewer that latency of inference is an important metric to evaluate the performance of our proposal. For this rebuttal, given the limited time, we would like to qualitatively discuss the performance of LO-BCQ. As demonstrated in Keller et al [3], with a dedicated inference accelerator for 64-wide group quantization, considerable speedup can be achieved, especially during the pre-fill phase in LLMs involving GEMM computations. In LO-BCQ, in addition to group quantization, we perform a decoding step where each 4-bit scalar is decoded to 6-bit integers. As a result of a small number of codebooks utilized by LO-BCQ (footprint = 0.19KB), one can potentially store them in shared memory of each SM resulting in a small look-up cost. A dedicated hardware design can amortize the cost of this lookup through spatial and temporal re-use of operands. We would like to emphasize that, as a result of a small number of codebooks (<= 16) in our work, the cost of this lookup is considerably small compared to other codebook-based proposals such as QuIP[4] and AQLM[5] where the number of codebooks are as large as 2^16.
> >
> > In response to the reviewer’s request for additional experimental results, we provide evaluations of LO-BCQ on Llama3.1-8B and Nemotron4-340B models and Wikitext-103 dataset. As shown, for a given group size, LO-BCQ achieves increasingly better perplexity as the number of codebooks are increased.
> >
> > # Nemotron4-340B: Perplexity (PPL) on Wikitext-103 (Baseline PPL = 3.48)
> >
> > | Num Codebooks | Group Size = 64 | Group Size = 32 | Group Size = 16 |
> > |---------------|-----------------|-----------------|-----------------|
> > | 2             | 3.67            | 3.63            | 3.61            |
> > | 4             | 3.62            | 3.61            | 3.58            |
> > | 8             | 3.60            | 3.59            | 3.57            |
> > | 16            | 3.59            | 3.56            | 3.55            |
> >
> > # Llama3.1-8B: Perplexity (PPL) on Wikitext-103 (Baseline PPL = 5.57)
> >
> > | Num Codebooks | Group Size = 64 | Group Size = 32 | Group Size = 16 |
> > |---------------|-----------------|-----------------|-----------------|
> > | 2             | 6.40            | 6.18            | 6.07            |
> > | 4             | 6.07            | 6.03            | 5.95            |
> > | 8             | 5.97            | 5.96            | 5.87            |
> > | 16            | 5.94            | 5.88            | 5.84            |
> >
> > We agree with the reviewer that additional experimental evaluations on MOE models would be helpful in fully demonstrating the benefits of our method. We will explore this as future work.

---

> ### Author Response · Authors · 2024-11-27
>
> Finally, we quantitively compare LO-BCQ to GPTQ[6], which is an effective method for 4-bit weight-only PTQ. Here, GPTQ quantizes weights to 4-bits with group-size of 128 while maintaining activations in 8-bits. In contrast, LO-BCQ quantizes both weights and activations to 4-bits with group size of 64. With >=8 codebooks and effective bitwidth of 4.5 and 4.625, respectively, LO-BCQ achieves lower perplexity loss compared to GPTQ despite quantizing activations to 4-bits. Further, we evaluate LO-BCQ on Wikitext-103, which is a significantly larger datatset compared to Wikitext-2 used in evaluating GPTQ. By demonstrating lower perplexity loss in larger dataset, we show that LO-BCQ is a significantly more efficient quantization method compared to GPTQ.
>
> # PPL Loss Compared to Unquantized Baseline
>
> | Model        | GPTQ (W4A8) | LO-BCQ (W4A4, 8 Codebooks) | LO-BCQ (W4A4, 16 Codebooks) |
> |--------------|-------------|----------------------------|-----------------------------|
> | LLAMA2-7B    | 0.22        | 0.13                       | 0.12                        |
> | LLAMA2-70B   | 0.10        | 0.09                       | 0.07                        |
>
> References:
> [1]: Rouhani et al, “Microscaling data formats for deep learning, 2023”
> [2]: Rouhani et al. “With shared microexponents, a little shifting goes a long way”. ISCA ’23
> [3]: Keller et al, “A 95.6-TOPS/W Deep Learning Inference Accelerator With Per-Vector Scaled 4-bit Quantization in 5 nm”. JSSC’23
> [4]: Chee et al, “QuIP: 2-Bit Quantization of Large Language Models With Guarantees, 2024
> [5]: Egiazarian et al, “Extreme compression of large language models via additive quantization, 2024.”
> [6]: Frantar et al, “GPTQ: Accurate Post-Training Quantization for Generative Pre-trained Transformers”

---

### Official Review · Reviewer_LF8w · 2024-11-02

**Soundness:** 2
**Presentation:** 2
**Contribution:** 2
**Rating:** 3
**Confidence:** 4

**Summary:**

The Paper introduces Block Clustered Quantization (BCQ), a novel method aimed at achieving accurate sub-8-bit quantization for weights and activations in large language models without relying on quantization-aware training (QAT). BCQ decomposes operand tensors into blocks, clusters these blocks and designs dedicated optimal quantization codebooks for each cluster. This approach allows for tailored quantization that minimizes mean squared error (MSE) and maintains high inference accuracy.

**Strengths:**

- Using LO-BCQ algorithm utilizing block clustering and dedicated codebooks effectively minimizes mean squared error (MSE) while maintaining high accuracy.
- The paper considers several datasets to evaluate, especially include the MMLU boolq and Race datasets.
- Achieving less than 1% loss when quantizing weights and activations to 4-bits.

**Weaknesses:**

- The experiment shows that LO-BCQ takes good performance but it is built on multiple codebooks and small block division, the current LLM has a huge number of parameters, 8 elements for a block division will lead to a large computational overhead, and it is still a question whether the good performance can be maintained if the size of the block is increased.

- LO-BCQ's fine-grained quantization does not seem to have an advantage over per-group quantization such as Atom[1], which quantizes 128 consecutive elements as a group, and achieves excellent results in W4A4, and even better if Atom's grouping is adjusted to 64 or smaller. In addition, LO-BCQ introduces operations such as clustering, indexing, updating codebooks, etc., which costs more computational overhead.
[1]Zhao, Yilong, et al. "Atom: Low-bit quantization for efficient and accurate llm serving.

**Questions:**

- In Table 4 Ablation experiments, only the case where $L_{b}$ is less than 8 is discussed, what is the performance of the model if the size of $L_{b}$ is aligned with the group quantization e.g. 128, 64?
- The performance improvement of LO-BCQ seems to partly come from the clustering of blocks, the value of activation in LLM has a large difference in the distribution of different datasets, does the use of pre-calibrate for activation to calculate the codebook of activation have an impact on the language modeling ability of LLM?
- What is the impact on model inference speed of performing clustering with block size 8, indexing, and looking up codebooks during model inference? Does it mask the gains from low-bit quantization?

---

> ### Author Response · Authors · 2024-11-25
>
> We thank the reviewer for their thorough comments and questions. We are glad to know that the reviewer acknowledges our extensive experimental evaluations across several LLMs and datasets. We also appreciate the reviewer emphasizing the fact that we have achieved <1% accuracy loss in several inference tasks when quantizing both weights and activations to 4-bits.
>
> The main concern of the reviewer is regarding the group-size or ‘block division’ of quantization. We agree with the reviewer that a small group-size such as 8 can lead to large computational overheads due to high precision scale factor multiplication per-group. We would like to clarify that the block-array size denoted by ‘$L_A$’ in our work is equivalent to group-size, that is, each block array shares a scale-factor. Therefore, in our work, scale factors are shared among groups of size 64, rather than 8, which has much smaller overhead. Indeed, one of the strengths of our work is using different granularities for scaling and codebook indexing (the latter which has size of 8). Therefore, LO-BCQ leverages the benefits of finer-grained quantization while avoiding a prohibitive overhead associated with scaling factors. We clarify this in more detail in part 1 of our response.
>
> We agree with the reviewer that Atom[1], a group-quantization based proposal, demonstrates impressive results across several inference tasks. We have now cited this work in our paper. With LO-BCQ, we demonstrate comparatively better inference accuracy than previous group quantization proposals across several benchmarks, including Atom[1] in part 2 of our response. The superior accuracy achieved by LO-BCQ can be attributed to flexibly mapping each block to a codebook that achieves least quantization error. In contrast, group quantization proposals associate a single number format to each group. Additionally, the reviewer has asked an important question regarding the calibration of codebooks for activations that are typically generated on-the-fly during inference. We have provided the performance of LO-BCQ on language modelling task with universally calibrated codebooks and compared against calibrating layerwise in part 3 of our response.
>
> Finally, the reviewer has raised valid concerns regarding the performance of our clustering procedure to identify the locally optimal codebooks. We would like to clarify that these steps are performed during the calibration phase (offline) and does not impact the latency of inference. We discuss this further in part 4 of our response.
>
> 1. Block-size ($L_b$) vs Block Array size ($L_A$):
> LO-BCQ shares a codebook per-$L_b$-element block and a scale-factor per-$L_A$-element block array. Here, block array is equivalent to “group size” in previous proposals such as Atom[1]. During inference, we perform $L_A$ wide inner dot-products which are scaled by the per-block array scale factors. Each 4-bit scalar within the block array is decoded into a 6-bit integer by looking up a 16-entry codebook. This codebook is selected by the codebook selector that is shared by each block of $L_b$ (typically = 8) elements within the block array. We would like to emphasize that, as a result of a small number of codebooks (<= 16) in our work, the cost of this lookup is considerably small compared to other codebook-based proposals such as QuIP#[4] and AQLM[5] where the number of codebooks are as large as $2^{16}$.
>
> 2.	Comparing LO-BCQ to other group quantization proposals:
> We present the inference accuracy achieved by LO-BCQ with a large group size of 64 across several datasets and models. Across these benchmarks, we demonstrate significantly better inference accuracy compared to other group quantization proposals such as Atom[1], MX[2] and MXFP[3].
> We find that with a group size of 64, LO-BCQ achieves significantly lower perplexity loss on Wikitext datatset compared to other works. We share an 8-bit scale-factor across 64 element block arrays, and each block of 8 shares a codebook selector. With 2, 4, 8 and 16 codebooks, the effective bit-width of LO-BCQ is 4.25, 4.375, 4.5 and 4.625, respectively.
> Similarly, we demonstrate significantly lower accuracy loss in datasets such as PIQA, BoolQ, HellaSwag and Winogrande. The superior performance of LO-BCQ is from effectively designing codebooks that minimize quantization error and flexibly mapping each block to a codebook that best quantizes it.

---

> ### Author Response · Authors · 2024-11-25
>
> ### LO-BCQ (Group Size=64): Loss compared to unquantized baseline (lower is better)
> | Num Codebooks | Model       |          Wiki-103 | PIQA  | BoolQ | HellaSwag | Winogrande |
> |---------------|-------------|-------------------|-------|-------|-----------|------------|
> | 2             | Llama2-7B   |          0.25     | 0.98  | 1.71  | 1.17      | 0.48       |
> |               | Llama2-70B  |          0.21     | 0.11  | 2.41  | 1.15      | 1.18       |
> | 4             | Llama2-7B   |          0.2      | 1.25  | 1.92  | 1.53      | 0.95       |
> |               | Llama2-70B  |          0.11     | 0.87  | 0.95  | 0.48      | 1.5        |
> | 8             | Llama2-7B   |          0.13     | 0.98  | 1.86  | 0.59      | -0.39      |
> |               | Llama2-70B  |          0.09     | 0.11  | 1.2   | 0.51      | 1.58       |
> | 16            | Llama2-7B   |          0.12     | 0.21  | 1.19  | 0.48      | 1.19       |
> |               | Llama2-70B  |          0.07     | 0.06  | 0.98  | 0.23      | 0.79       |
>
> ### ATOM[1] (Group Size = 128): Loss compared to unquantized baseline (lower is better)
> | Model      | Wiki2 | PIQA  | BoolQ | HellaSwag | Winogrande |
> |------------|-------|-------|-------|-----------|------------|
> | LLAMA-7B   | 0.48  | 1.09  | 3.33  | 3.18      | 3.16       |
> | LLAMA-65B  | 0.36  | 0.38  | 0.24  | 1.61      | 4.5        |
>
> ### MX4[2]  (Group Size = 16): Loss compared to unquantized baseline (lower is better)
> | Model      | Wiki2 | PIQA  | BoolQ | HellaSwag | Winogrande |
> |------------|-------|-------|-------|-----------|------------|
> | LLAMA-7B   | 0.67  | 1.03  | 5.31  | 1.91     | 3.16       |
> | LLAMA-65B  | 0.44  | 0.98  | 2.82  | 2.03      | 3.55     |
>
> ### MXFP4[3]  (Group Size = 32): Loss compared to unquantized baseline (lower is better)
> | Model      | Wiki2 | PIQA  | BoolQ | HellaSwag | Winogrande |
> |------------|-------|-------|-------|-----------|------------|
> | LLAMA-7B   | 0.70  | 0.54  | 5.29  | 2.88      | 1.90       |
> | LLAMA-65B  | 0.54  | 0.98  | 2.17  | 2.03      | 3.63       |
>
> 3.	Impact on language-modeling task (perplexity on Wikitext-103) with universally calibrated codebooks:
> We calibrate our codebooks on activations from GPT3-126M model and freeze the codebooks across our evaluations. As demonstrated by the perplexity achieved by LO-BCQ on Wikitext-103 dataset, the pre-calibrated codebooks are effective in quantizing activations of various models we considered.
>
> ### Wikitext-103, Group size=64 : Loss compared to unquantized baseline (lower is better)
> | NUM CODEBOOKS | Llama2-7B | Llama2-70B | Nemotron4-15B | Nemotron4-340B | GPT3-22B |
> |---------------|-----------|------------|---------------|----------------|----------|
> | 2             | 0.25      | 0.21       | 0.43          | 0.19           | 0.20     |
> | 4             | 0.20      | 0.11       | 0.33          | 0.14           | 0.10     |
> | 8             | 0.13      | 0.09       | 0.26          | 0.12           | 0.08     |
> | 16            | 0.12      | 0.07       | 0.21          | 0.11           | 0.09     |
>
> To validate this further, we compare the perplexity achieved by codebooks calibrated individually for each operand (weights and activations) in each layer of Llama2-7B. As shown, the universally calibrated codebooks perform as well as layer-wise calibrated codebooks especially with >= 8 codebooks.
>
> ### Wikitext-103, Group size=64, Llama2-7B: Loss compared to unquantized baseline (lower is better)
> | NUM CODEBOOKS | Universally Calibrated | Layerwise Calibrated |
> |---------------|------------------------|----------------------|
> | 2             | 0.25                   | 0.23                 |
> | 4             | 0.20                   | 0.16                 |
> | 8             | 0.13                   | 0.13                 |
> | 16            | 0.12                   | 0.11                 |
>
> 4.	Impact of clustering, indexing and look-up steps of LO-BCQ on inference performance:
>
> We agree with the reviewer that the clustering and identifying codebooks is an expensive procedure and will impact inference performance. However, we would like to clarify that the clustering and calibration of codebooks is performed prior to inference (offline). We freeze the codebooks identified by LO-BCQ algorithm across our evaluations.
>
> Further, the codebook selectors are pre-determined for the weights and are computed on-the-fly for activations. The cost of determining the codebook selector is amortized by the large size of dot-product dimension (e.g. 8192 in Llama2-70B).
>
> Finally, each 4-bit scalar is decoded to 6-bit integer during GEMM/GEMV computations. As a result of a small number of codebooks utilized by LO-BCQ (footprint = 0.19KB), one can potentially store them in shared memory of each SM resulting in a small look-up cost. A dedicated hardware design can amortize the cost of this lookup through spatial and temporal re-use of operands.

---

> > ### Author Response · Authors · 2024-11-25
> >
> > References:
> > [1]: Zhao, Yilong, et al. "Atom: Low-bit quantization for efficient and accurate llm serving.
> > [2]: Rouhani et al. “With shared microexponents, a little shifting goes a long way”. ISCA ’23
> > [3]: Rouhani et al, “Microscaling data formats for deep learning, 2013”
> > [4]: Tseng et al, “Quip#: Even better llm quantization with hadamard incoherence and lattice codebooks, 2024
> > [5]: Egiazarian et al, “Extreme compression of large language models via additive quantization, 2024.”

---

> ### Author Response · Authors · 2024-12-01
>
> Additionally, with a group size of 128 for LO-BCQ, we would like to present a comparison to Atom. Here, LO-BCQ with $2$, $4$, $8$ and $16$ codebooks result in an effective bitwidth of $4.19$, $4.31$, $4.44$ and $4.56$, respectively, for both weights and activations.
>
> ## Wikitext PPL Loss compared to unquantized baseline
>
> | Model         | Atom | LO-BCQ (2 codebooks) | LO-BCQ (4 codebooks) | LO-BCQ (8 codebooks) | LO-BCQ (16 codebooks) |
> |---------------|------|-----------------------|-----------------------|-----------------------|------------------------|
> | **LLAMA2-7B** | 0.48 | 0.32                  | 0.21                  | 0.15                  | 0.15                   |
> | **LLAMA2-70B**| 0.36 | 0.29                  | 0.15                  | 0.12                  | 0.10                   |

---

### Official Review · Reviewer_miqq · 2024-11-05

**Soundness:** 4
**Presentation:** 3
**Contribution:** 3
**Rating:** 8
**Confidence:** 4

**Summary:**

The paper presents Block Clustered Quantization (BCQ), a post-training quantization (PTQ) method designed to compress both weights and activations in large language models (LLMs) to 4-bit precision. BCQ divides tensors into small blocks, clusters these blocks based on their statistical properties, and then quantizes each cluster using a dedicated codebook, reducing memory footprint and computational load while preserving model accuracy. The Locally Optimal Block Clustered Quantization (LO-BCQ) algorithm iteratively refines clusters and codebooks to minimize quantization error. With solid experiments with varying configurations on Llama2 and GPT3 models,  it has shown LO-BCQ outperforms the states-of-the-art methods on W4A4 quantization.

**Strengths:**

1. Introducing iterative block clustering to derive an optimal quantization codebook for each block is novel, providing a robust and flexible mapping to the most accurate representation of each block. This approach, I believe, is the core factor driving the success of this method.

2. It is a compelling finding that the codebook is universally applicable for quantizing any tensor, at any layer, across different models.

**Weaknesses:**

I find this paper impressive, with no obvious weaknesses in its methodology or presentation. Please refer to the Questions section for some discussion points

**Questions:**

1. How does this method perform on W2A2 and W8A8 quantization, and how does it compare to other methods? What are the bottlenecks when applying this method to more aggressive quantization?

    2. As shown in Fig. 6, the range of values within some codebooks is smaller than [-1, 1], which appears to compensate for the large L_A. Have you tested even smaller L_A values, and how do they perform?

    3. 223: fond → found.

---

> ### Author Response · Authors · 2024-11-27
>
> We greatly appreciate the comments and questions of the reviewer. We thank the reviewer acknowledging that our codebook-based approach is universally applicable for quantizing any tensor, at any layer, across different models.
>
> We have conducted some preliminary studies with LO_BCQ for sub-4-bit quantization of weights (W2A4 quantization). In this regime, we find that quantization-aware training is needed to bridge the accuracy gap from unquantized baseline. Further, reducing the block array size ($L_A$) and block size ($L_b$) result in improved accuracy at the cost of increased effective bit-width. This is because at smaller $L_A$ and $L_b$, the per-block array scale-factors and per-block codebook selectors are shared among fewer scalars.
>
> The reviewer is correct is pointing out that when $L_A$ is large, the range of values within some codebooks are smaller than [-1,1] and these codebooks are used primarily for blocks with relatively small values. Although we have not experimented with $L_A$ smaller than 16, we would like to discuss the general trends. We find that the inference accuracy improves as $L_A$ decreases at the cost of increased bit-width overhead for storing the per-block array scale factors. Similarly, the inference accuracy improves when $N_c$ (number of codebooks) increases at the cost of increased bit-width overhead for storing the per-block codebook selectors. Empirically, we find that it is worthwhile to increase the number of codebooks $N_c$ rather than decrease $L_A$. For, instance the table below reports the Wikitext-103 perplexity on Llama2-70B model with various $L_A$ and $N_c$ values for $L_b$=8. With $L_A$=64, increasing the number of codebooks to 16 results in better perplexity (lower is better), compared to decreasing $L_A$ to 16, where the effective bit-width in both these configurations is 4.625.
>
> ## Wikitext-103 perplexity on Llama2-70B model
> | \( N_c ) | 2    | 4    | 8    | 16   |
> |-----------|------|------|------|------|
> | \( L_A = 64 \) | 3.35 | 3.25 | 3.23 | 3.21 |
> | \( L_A = 32 \) | 3.27 | 3.24 | 3.22 | 3.20 |
> | \( L_A = 16 \) | 3.25 | 3.22 | 3.20 | 3.19 |

---

### Author Response · Authors · 2024-11-27

We would like to thank all the reviewers for their insightful comments and questions. We appreciate the interest in codebook-based approach for achieving 4-bit weight and activation quantization in LLMs. We would like to emphasize that to the best of our knowledge, we are the first to achieve < 1% loss in downstream task accuracy when both LLM activations and weights are quantized to 4-bits during PTQ (no finetuning).

## Key Modifications:
The main criticism of our work is regarding the group-size (referred as block array size in our paper) of quantization. We agree with the reviewers that a small group-size such as 8 can lead to large computational overheads due to high precision scale factor multiplication per-group. However, we would like to clarify that the block-array size denoted by ‘$L_A$’ in our work is equivalent to group-size, that is, each block array shares a scale-factor. Therefore, in our work, scale factors are shared among groups of size 64, rather than 8, which has much smaller overhead.

In response to reviewers’ suggestion to compare inference accuracy of LO-BCQ against other weight-only quantization proposals, we report perplexity loss on Wikitext dataset during weight-only LO-BCQ. We explore a large group-size of 128 and show that LO-BCQ achieves lower perplexity loss compared to other methods.  Here, the LO-BCQ configuration is denoted by tuple $\{L_A, L_b, N_c\}$ = {Length of block array, Length of block, Number of codebooks}. LO-BCQ with $2$, $4$, $8$ and $16$ codebooks result in an effective bitwidth of $4.19$, $4.31$, $4.44$ and $4.56$, respectively.

### Wikitext perplexity loss compared to unquantized baseline
| Method                      | Llama2-7B | Llama2-70B |
|-------------|-----------|------------|
| GPTQ                        | 0.22     | 0.10      |
| AWQ                         | 0.13     | **0.09**  |
| OmniQ                       | 0.27     | 0.15      |
| QuiP#                       | 0.19     | 0.10      |
| **LO-BCQ** \{128,8,2\} | 0.14     | **0.09**  |
| **LO-BCQ** \{128,8,4\} | 0.12     | **0.07**  |
| **LO-BCQ** \{128,8,8\} | **0.09** | **0.06**  |
| **LO-BCQ** \{128,8,16\} | **0.08** | **0.05**  |

The superior performance of LO-BCQ is from effectively designing codebooks that minimize quantization error and flexibly mapping each block to a codebook that best represents it.

## Other modifications
- Figure1: We include comparison to Atom which is an excellent group quantization-based proposal for quantizing both weights and activations in LLMs-
- Table 3. (b): We report weight-only quantization results for LO-BCQ with a large group size of 128 and compare it to GPTQ, AWQ, OmniQ and QuiP#.
- Table 7: We report Wikitext-103 perplexity results with Nemotron4-340B model and various LO-BCQ configurations
- We fixed the spelling errors pointed out by the reviewers.

---

### Meta-Review · Area_Chair_JKrK · 2024-12-20

**Metareview:**

The paper presents a novel quantization technique, Block Clustered Quantization (BCQ), aimed at significantly reducing the storage and computational needs of large language models (LLMs) while maintaining near-state-of-the-art inference accuracy. This is achieved through the decomposition of operand tensors into blocks, clustering these blocks based on statistical characteristics, and employing tailored quantization codebooks for each cluster. The authors argue that the proposed method is innovative and addresses the critical challenge of achieving efficient sub-8-bit quantization without further training costs.

The reviewers exhibited varied perspectives on the paper’s merits. While some acknowledged the algorithm's technical novelty and its potential for improving quantization accuracy, concerns were raised regarding the lack of clarity in differentiating BCQ from existing group-wise quantization methods. Notably, one reviewer recognized the paper's innovative approach yet categorized it as marginally below the acceptance threshold, indicating mixed opinion among the reviewers.

Despite receiving a rebuttal from the authors, significant concerns remained unaddressed, particularly surrounding inference latency. While the authors asserted that their method improves inference accuracy, several reviewers questioned whether the added complexity of block clustering and codebook indexing could offset these gains, particularly in practical applications. One reviewer emphasized that the proposed method might not deliver substantial improvements in inference speed, echoing concerns that persisted beyond the authors' responses.

In conclusion, while the paper has merit and could contribute to the field of quantization in LLMs, the unresolved issues regarding inference latency and the clarity of its novelty compared to existing methods hinder unanimous support for acceptance. Based on the reviewers' insights, the paper does not currently meet the criteria for acceptance due to these outstanding concerns.

**Additional Comments On Reviewer Discussion:**

The reviewers exhibited varied perspectives on the paper’s merits. While some acknowledged the algorithm's technical novelty and its potential for improving quantization accuracy, concerns were raised regarding the lack of clarity in differentiating BCQ from existing group-wise quantization methods. Notably, one reviewer recognized the paper's innovative approach yet categorized it as marginally below the acceptance threshold, indicating mixed opinion among the reviewers.

Despite receiving a rebuttal from the authors, significant concerns remained unaddressed, particularly surrounding inference latency. While the authors asserted that their method improves inference accuracy, several reviewers questioned whether the added complexity of block clustering and codebook indexing could offset these gains, particularly in practical applications. One reviewer emphasized that the proposed method might not deliver substantial improvements in inference speed, echoing concerns that persisted beyond the authors' responses.

---

### Decision · Program_Chairs · 2025-01-22

Reject